# Impacts of global NO$_x$ inversions on NO$_2$ and ozone simulations

Zhen Qu[1,2], Daven K. Henze[1], Owen R. Cooper[3,4], Jessica L. Neu[5]

[1]Department of Mechanical Engineering, University of Colorado Boulder, Boulder, CO, 80309, USA
[2]School of Engineering and Applied Science, Harvard University, Cambridge, MA, 02138, USA
[3]Cooperative Institute for Research in Environmental Sciences, University of Colorado, Boulder, CO, 80309, USA
[4]NOAA Chemical Sciences Laboratory, Boulder, CO, 80305, USA
[5]Jet Propulsion Laboratory, California Institute of Technology, Pasadena, CA, 91109, USA

*Correspondence to*: Zhen Qu (zhen.qu@colorado.edu)

**Abstract.** Tropospheric NO2 and ozone simulations have large uncertainties, but their biases, seasonality and trends can be improved with NO2 assimilations. We perform global top-down estimates of monthly NO$_x$ emissions using two OMI NO$_2$ retrievals (NASAv3 and DOMINOv2) from 2005 to 2016 through a hybrid 4D-Var / mass balance inversion. Discrepancy in NO$_2$ retrieval products is a major source of uncertainties in the top-down NO$_x$ emission estimates. The different vertical sensitivities in the two NO$_2$ retrievals affect both magnitude and seasonal variations of top-down NO$_x$ emissions. The 12-year

averages of regional NO$_x$ budgets from the NASA posterior emissions are 37% to 53% smaller than the DOMINO posterior. Consequently, the DOMINO posterior surface NO$_2$ simulations greatly reduced the negative biases in China (by 15%) and the US (by 22%) compared to surface NO$_2$ measurements. Posterior NO$_x$ emissions show consistent trend over China, US, India, and Mexico constrained by the two retrievals. Emission trends are less robust over South America, Australia, Western Europe and Africa, where the two retrievals show less consistency. NO$_2$ trends have more consistent decreases (by 26%) with the

measurements (by 32%) in the US from 2006 to 2016 when using the NASA posterior. The performance of posterior ozone simulations has spatial heterogeneities from region to region. On a global scale, ozone simulations using NASA-based emissions alleviates the double peak in the prior simulation of global ozone seasonality. The higher abundances of NO$_2$ from the DOMINO posterior increase the global background ozone concentrations and therefore reduce the negative biases more than the NASA posterior in the GEOS-Chem v12 simulations at remote sites. Compared to surface ozone measurements,

posterior simulations have more consistent magnitude and interannual variations than the prior estimates, but the performance from the NASA-based and DOMINO-based emissions varies across ozone metrics. The limited availability of remote sensing data and the use of prior NO$_x$ diurnal variations hinder improvement of ozone diurnal variations from the assimilation, and therefore have mixed performance on improving different ozone metrics. Additional improvements in posterior NO$_2$ and ozone simulations require more precise and consistent NO$_2$ retrieval products, more accurate diurnal variations of NO$_x$ and VOC

emissions, and improved simulations of ozone chemistry and depositions.

## 1 Introduction

Tropospheric ozone is a harmful secondary air pollutant affecting human health, sensitive vegetation, and ecosystems [NRC, 1991; Monks et al., 2015]. Long-term ozone ($O_3$) exposure is estimated to cause $1.04 - 1.23$ million respiratory deaths in adults [Malley et al., 2017]. Short-term exposure to high ambient ozone is associated with respiratory and cardiovascular mortality [Turner et al., 2016; Fleming et al., 2018]. Accurate simulations of ozone in highly polluted regions are important for better pollution forecasts and more effective emission regulations. Tropospheric ozone is formed through photochemical reactions between nitrogen oxide ($NO_x = NO + NO_2$), carbon monoxide (CO), methane ($CH_4$), and volatile organic compounds (VOCs) in the presence of sunlight [Crutzen, 1973; Derwent et al., 1996]. These precursor gases are mainly emitted from fossil fuel combustion, biomass burning, oil and gas production, industry, agriculture, and biogenic activities. Tropospheric ozone can also be transported from the stratosphere through stratosphere-troposphere exchange [Stohl et al., 2003; Hsu and Prather, 2009; Stevenson et al., 2006], but this magnitude is smaller than the amount from chemical production by a factor of $5 - 7$ [Young et al., 2013]. Ozone is removed from the troposphere through deposition [Fowler et al., 2009], photo-dissociation, and reactions with $HO_2$, $NO_2$, unsaturated VOCs, halogens, and aerosols [Crutzen, 1973].

From 1850 to 2000, global mean tropospheric ozone burden has increased by 29% [Young et al., 2013]. Human activities are major sources of ozone precursor gases, contributing to 9% (24.98 Tg) increase of the global tropospheric ozone burden from 1980 to 2010 [Zhang et al., 2016]. Ozone formation and trends depend nonlinearly on the local relative abundances of $NO_x$ and VOCs and the radiative regime in which these occur. Previous studies have shown that changes in surface ozone are dominated by regional emission trends of precursor gases [Zhang et al., 2016]. At the global scale, 77% of $NO_x$ emissions are from anthropogenic sources, according to the HTAP 2010 inventory [Janssens-Maenhout, 2015]. Anthropogenic $NO_x$ emissions have been decreasing in North America and Europe due to transportation and energy transformations [Simon et al., 2015], but have been increasing in China up until 2011 according to bottom-up emission inventories [Liu et al., 2016; Hoesly et al., 2018]. Top-down $NO_x$ emission estimates using satellite observations from the Ozone Monitoring Instrument (OMI) showed a similar turning point in China [Miyazaki et al., 2017; Qu et al., 2017], but a slowdown in reductions in the US compared to bottom-up estimates [Miyazaki et al., 2017; Jiang et al., 2018]. However, in India and the Middle East, where ozone production is more efficient than higher latitude regions [Zhang et al., 2016], $NO_2$ column densities from OMI are continuing to increase [Krotkov et al., 2016].

Top-down methods have the advantage of being able to update emissions in a more timely fashion than the bottom-up approaches; still, top-down approaches can contain large differences and uncertainties. For instance, the magnitude of tropospheric $NO_2$ column densities from two global retrievals from the National Aeronautics and Space Administration (NASA) and the Royal Netherlands Meteorological Institute (KNMI) differ by 50%, and have different trends at the regional scale [Zheng et al., 2014; Canty et al., 2015; Qu et al., 2017]. These differences in column densities can propagate to differences in

top-down $NO_x$ emission estimates [e.g., Miyazaki et al., 2017; Qu et al., 2017]. In this study, we assess the importance of these discrepancies in $NO_x$ emissions for the simulation of ozone. We derive global top-down $NO_x$ emissions from 2005 to 2016 using two widely used products (OMNO2 v3 and Dutch OMI $NO_2$ (DOMINO) v2) based on the same inversion process for consistent evaluations (Sect. 3). We also evaluate a new OMI $NO_2$ retrieval product, the Quality Assurance for the Essential Climate Variables (QA4ECV) [Boersma et al., 2018], and apply it to derive monthly $NO_x$ emissions in 2010. We do not repeat

our entire set of ozone evaluations with this product given that its magnitude and seasonality does not significantly differ from the other two products. We further explore the impact of adjusting $NO_x$ emissions on ozone simulations, by evaluating the ozone simulations produced from bottom-up and top-down $NO_x$ emissions against global surface measurements from the Tropospheric Ozone Assessment Report (TOAR) database and the China National Environmental Monitoring Center (CNEMC) network.


In addition to local sources, the lifetime of ozone (~22 days on global average) is sufficiently long enough for intercontinental transport [UNECE, 2010]. Consequently, every country is an exporter as well as an importer of ozone pollution. Transport from East Asia can be an important contributor to ozone exceedances in the western US [Goldstein et al., 2004; Zhang et al., 2009; Zhang et al., 2014; Fiore et al., 2014; Verstraeten et al., 2015; Lin et al., 2017; Jaffe et al., 2018]. The influence of

intercontinental ozone transport is strongest in spring and summer, when background ozone concentrations reach 50 ppbv at the west coast of the US [Jaffe et al., 2018]. The impact of background ozone is increasingly important and challenging due to the decreased local sources of precursor gases in the US [Hoesly et al., 2018] and the recent stricter ozone standard in the US lowering the annual 4[th] highest maximum daily 8-hour average ozone concentration from 75 ppbv to 70 ppbv in 2015 [Cooper et al., 2015]. Optimization of $NO_x$ emissions in the upwind regions can improve remote ozone simulations in downwind regions

after transport of intercontinental pollution plumes from the free troposphere to the surface [Zhang et al., 2008; Verstraeten et al., 2015]. Therefore, we also evaluate the model simulations of remote ozone at the west coast of the United States using bottom-up and top-down $NO_x$ emissions in Sect. 4.

## 2 Methods

### 2.1 GEOS-Chem and its adjoint model

The GEOS-Chem adjoint model [Henze et al., 2007] v35k is used to derive global $NO_x$ emission estimates at $2° \times 2.5°$ resolution. It was developed for inverse modelling of aerosol and gas emissions using the 4D-Var method by Henze et al. [2007, 2009] based on version 8 of GEOS-Chem, with bug fixes and updates up to version 10. Simulations in this study are driven by Modern-Era Retrospective analysis for Research and Applications, Version 2 (MERRA-2) meteorological fields from NASA Global Modeling and Assimilation Office (GMAO). Anthropogenic emissions of $NO_x$, $SO_2$, $NH_3$, CO,

NMVOCs and primary aerosol from the HTAP 2010 inventory version 2 [Janssens-Maenhout et al., 2015] are used to drive all prior simulations from 2005 to 2017. The diurnal variation of $NO_x$ emissions is derived from EDGAR hourly variations (

http://wiki.seas.harvard.edu/geos-chem/index.php/Scale_factors_for_anthropogenic_emissions#Diurnal_Variation) and is not optimized in the inversion. The use of non-anthropogenic emissions and other setups follow Qu et al. [2017, 2019]. In the following analyses, we refer to this model as "GC-adj."


GC-adj does not include several halogen chemistry mechanisms that affect ozone depletions primarily over the oceans [Sherwen et al., 2016a; Wang et al., 2019] and at high altitude regions [Sherwen et al., 2016a]. Given their impact on the global background ozone concentrations, we also use GEOS-Chem v12.1.1 to evaluate ozone simulations at $2° \times 2.5°$ resolution driven by the MERRA-2 meteorological fields. The chemistry updates include the stratospheric chemistry from the Universal

tropospheric-stratospheric Chemistry eXtension (UCX) [Eastham et al., 2014], halogen chemistry [Bell et al., 2002; Parrella et al., 2012; Sherwen et al., 2016a, 2016b; Schmidt et al., 2016; Sherwen et al., 2017], and updated isoprene and monoterpene chemistry [Chan Miller et al., 2017; Fisher et al., 2016; Marais et al., 2016; Travis et al., 2016]. The Harvard-NASA Emission Component (HEMCO) is employed to process emissions in this version of GEOS-Chem [Keller et al., 2014]. We use 72 levels of vertical grid and global anthropogenic emissions from the Community Emissions Data System (CEDS) [Hoesly et al., 2018].

Top-down $NO_x$ emissions derived using GC-adj are also input to this model to evaluate the impact of $NO_2$ data assimilation on ozone simulations under different chemical mechanisms. We refer to this model as "GCv12" in this manuscript.

For each $NO_x$ emission dataset, the model is spun-up for 6 months, starting from July 2005. Therefore, we derive $NO_x$ emissions from 2005, but only evaluate simulations with measurements from 2006. To avoid high biases when comparing

simulated ozone averaged over the first vertical model layer (~100 m in box height) with surface measurements, 2-meter ozone mixing ratios are calculated by scaling simulated ozone mixing ratios in the first layer using adjusted dry deposition velocities at 2 meters following Zhang et al. [2012] and Lapina et al. [2015].

**2.2 Satellite observations and global top-down $NO_x$ emissions**

We estimate global top-down $NO_x$ emissions at the surface from 2005 to 2016 at $2° \times 2.5°$ resolution using tropospheric $NO_2$

column densities from OMI. OMI is an Ultraviolet/Visible nadir solar backscatter spectrometer aboard the NASA Aura satellite. It has a local overpass time of about 13:45 and a nadir resolution of 13 km × 24 km. OMI was launched in July 2004 and has provided operational data products since October 2004. Two Level 2 $NO_2$ retrieval products are used to derive long-term top-down $NO_x$ emissions in this study: the NASA standard product OMNO2 version 3 [Krotkov et al., 2017] and the DOMINO version 2 from KNMI [Boersma et al., 2011]. A new OMI $NO_2$ retrieval, the Quality Assurance for the Essential Climate

Variables (QA4ECV) [Boersma et al., 2018], has recently become available. This product is jointly developed by KNMI, the Belgian Institute for Space Aeronomy (BIRA-IASB), University of Bremen, Max-Plank Institute for Chemistry, and Wageningen University. We evaluate the magnitude of $NO_2$ column densities and the seasonality of posterior $NO_x$ emissions in 2010 from this product. We screen all OMI $NO_2$ retrievals using data quality flags and by the criteria of positive tropospheric

column, cloud fraction < 0.2, solar zenith angle < 75°, and viewing zenith angle < 65°. We excluded all retrievals that are

affected by row anomaly.

We converted GEOS-Chem $NO_2$ VCD to SCD using scattering weight from the OMI retrievals and then compare GEOS-Chem SCD with SCD retrieved from OMI. The scattering weights are the product of the averaging kernels and the air mass factor (AMF) [Palmer et al., 2001; Chance and Martin 2017]. A cost function is defined as the observation error weighted

differences between simulated and retrieved $NO_2$ SCD, plus the prior error weighted departure of the emission scaling factors from the prior estimates. We minimize the cost function using the quasi-Newton L-BFGS-B gradient-based optimization technique [Byrd et al., 1995; Zhu et al., 1994], in which the gradient of the cost function with respect to the control parameter is calculated using the adjoint method. Details of the assimilation of $NO_2$ slant column densities (SCDs), how vertical sensitivities of satellite retrievals are accounted for, and the hybrid 4D-Var / mass balance inversion of $NO_x$ emissions are

described in Qu et al. [2017]. We use top-down $NO_x$ emissions estimated from the NASA standard product and the DOMINO product in the evaluations of ozone simulations.

### 2.3 Surface measurements

We evaluate surface $NO_2$ simulations with measurements from the Environmental Protection Agency (EPA) Air Quality System (AQS) in the US and the China National Environmental Monitoring Center (CNEMC) network in China. The city

monitoring sites included in the analysis represent either urban background or the averaged pollutant concentrations over the city. Simulated ozone mixing ratios from 2006 to 2016 are compared to surface observations from the TOAR Surface Ozone Database [Schultz et al., 2017] at the global scale and the CNEMC network in China. TOAR has produced a relational database of global surface ozone observations at all available sites; see Gaudel et al. [2018] for illustrations of the global coverage of the TOAR data. Precompiled TOAR data (https://doi.pangaea.de/10.1594/PANGAEA.876108, available from 1995 to 2014)

at each individual site are used in this study. Given the sparse TOAR data coverage of only 32 sites over China, hourly surface ozone measurements from the CNEMC (http://106.37.208.233:20035/) are used to evaluate simulations in China from 2014 to 2016. The CNEMC national network was designed for urban and suburban air pollution monitoring. The archive contains hourly observations of ozone, carbon monoxide, nitrogen dioxide, sulfur dioxide and fine particulate matter across mainland China since 2013.

### 155 2.4 Ozonesonde measurements

Ozone profile measurements from the Intercontinental Chemical Transport Experiment Ozonesonde Network Study (IONS-2010) [Cooper et al., 2011] are used to evaluate the continental inflow of ozone along the west coast of the United States, where air masses are not influenced by recent US emissions. IONS-2010 was a component of the California Research at the Nexus of Air Quality and Climate Change (CalNex) 2010 experiment [Ryerson et al., 2013] and was a continuation of previous

IONS experiments to measure tropospheric ozone variability across North America [Thompson et al., 2007, 2008; Cooper et

al., 2007]. Balloon-borne electrochemical cell sensors were used to measure ozone profiles with an accuracy of +/- 10% in the troposphere [Johnson et al., 2002; Smit et al., 2007]. All six sites in California from IONS-2010 (Trinidad Head, Point Reyes, Point Sur, San Nicolas, Joshua Tree, and Shasta) are included in this study. These measurements are made in the mid-afternoon (95% occurring between 14:00 and 16:59 local time) over a six-week period from May 10 to June 19, 2010. There are 34-37 profiles for all sites except for San Nicolas Island, where only 26 profiles are available due to multiple instrument failures. Measurements made between 700 – 800 hPa are used to evaluate remote ozone simulations.

## 3 Magnitude, seasonality and trend of $NO_x$ emissions, surface $NO_2$ and surface ozone

Differences between the prior and posterior $NO_x$ emission estimates are mainly driven by the differences between simulated and retrieved tropospheric $NO_2$ vertical column densities (VCDs), which are compared in Sect. S1 in the supporting information. The GEOS-Chem $NO_2$ SCDs converted using scattering weight from the NASA product are larger than the SCDs calculated using the DOMINO scattering weight and the same GEOS-Chem VCDs (See Fig. S2). These can be explained by the use of different surface albedo and cloud product in the two retrievals. The retrieved $NO_2$ SCDs from the NASA product are mostly smaller than the DOMINO retrieval except for some regions between 40°N – 60°N in January 2010. The smaller magnitude in OMI SCD and the larger magnitude in GEOS-Chem SCD using the NASA scattering weight lead to smaller magnitude of posterior $NO_x$ emissions than inversions from the DOMINO product. The cost function has reduced by 6% - 29% in the monthly inversion.

### 3.1 Annual average

As shown in Table 1, the global budgets of $NO_x$ emissions from the NASA posterior in 2010 is 0.7% smaller than the prior; DOMINO posterior is 18% larger than the prior; QA4ECV posterior is 11% larger than the prior. The positive increment in the DOMINO posterior emissions is consistent with the +26% increments of 10-year mean posterior $NO_x$ emissions in Miyazaki et al. [2017]. The annual global $NO_x$ emissions from Miyazaki et al. [2017] are between 46.7 Tg N yr$^{-1}$ and 50.9 Tg N yr$^{-1}$ from 2005 to 2014, which are within 31% from the DOMINO posterior emissions in 2010 in this study.

As shown in Fig. 1, the NASA posterior $NO_x$ emissions are less than the prior $NO_x$ emissions in the northeast US, northeast China, and southeast China. The DOMINO posterior $NO_x$ emissions are larger than the prior in most regions except for North Mexico and most parts of the tropics. The QA4ECV posterior $NO_x$ emissions have more consistent negative increments in Eastern China with the NASA posterior emissions and more consistent positive increments in the United States, India, Europe, and Australia with the DOMINO posterior emissions. At the regional scale, NASA posterior increments are -3% in China, -1% in the US, +0.3% in India, and -1% in Western Europe. The increments from the DOMINO posterior emissions are +21% in China, +31% in the US, +28% in India, and +38% in Western Europe. The different changing directions in the above two posterior $NO_x$ emissions are consistent with the reportedly higher magnitude of $NO_2$ column densities in the DOMINO product

than the NASA product in densely populated and industrial regions [Zheng et al., 2014; Canty et al., 2015; Qu et al., 2017]. The increments from the QA4ECV posterior emissions are +5% in China, +19% in the US, +18% in India, and +14% in Western Europe.


To evaluate the magnitude of the posterior $NO_x$ emissions, we compare simulations of surface $NO_2$ concentrations using the NASA and DOMINO based $NO_x$ emissions with surface measurements in the US and China. Surface $NO_2$ simulations at coarse resolution are usually biased low compared to measurements at urban sites, due to the short lifetime of $NO_x$. We therefore start with analysing this resolution error by generating high-resolution pseudo surface measurements at $0.1° \times 0.1°$

and compare them with low-resolution model simulations at $2° \times 2.5°$. We generate high-resolution surface $NO_2$ concentrations by scaling simulated surface $NO_2$ concentrations at $2° \times 2.5°$ grid cells by the ratio of OMI $NO_2$ column density gridded at $0.1° \times 0.1°$ to the OMI $NO_2$ column density gridded at $2° \times 2.5°$ grid cell. We identify $0.1° \times 0.1°$ grid cells that include surface monitoring sites and treat downscaled surface $NO_2$ concentrations at these grid cells as the pseudo surface measurements. Comparisons of pseudo surface measurements and $NO_2$ simulations at $2° \times 2.5°$ purely reflect differences caused by comparing

$NO_2$ concentrations at $2° \times 2.5°$ with higher resolution surface measurements at urban regions. The mean of the pseudo $NO_2$ measurements is 32% higher than the low-resolution simulations in the US, and it is 18% higher than the low-resolution simulations in China. The real surface measurements, which represent a single point within the $0.1° \times 0.1°$ grid cell, are expected to have even larger biases than the values calculated here, where we assume the measurements are at $0.1° \times 0.1°$ grid cells. The smaller bias in China in comparison to the US is related to the higher background $NO_2$ concentrations in China.


Figure 2 shows the comparisons of annual mean surface $NO_2$ concentrations in 2015 from measurements and simulations using different $NO_x$ emission inputs. The selection of this year is due to the limited availability of nation-wide surface $NO_2$ measurements in China. Surface $NO_2$ concentrations in both China and the US are measured by chemiluminescence analyzers, each equipped with a molybdenum converter, which converts additional $NO_y$ compounds to NO and leads to a positive bias in

$NO_2$ measurements [Dunlea et al., 2007; Steinbacher et al., 2007]. We therefore calculate a correction factor following Lamsal et al. [2008] for each GEOS-Chem simulation and divide the simulated $NO_2$ concentrations by this correction factor to convert simulated $NO_2$ to the measured species. The correction factors are generally higher in the US than in China, but have similar seasonality (see Fig. S3). Subtracting the resolution bias from the statistics shown on Fig. 2, the equivalent normalized mean bias (NMB) of surface $NO_2$ concentrations using the NASA posterior is -54% in China and -41% in the US. The equivalent

NMB using the DOMINO posterior is -38% in China and -19% in the US. These remaining negative biases reflect the unrepresentativeness of $0.1°$ pseudo measurements for real point measurements for resolution bias correction, comparison of $NO_2$ concentrations averaged over $2°×2.5°$ simulation to limited measurements, the underestimates of $NO_2$ retrievals using coarse resolution a priori, and the inability of data assimilation to increase emissions at grid cell where $NO_2$ retrievals are below the detection limit of OMI. Although we have not performed a $NO_x$ emission inversion using the QA4ECV product for

2015, we expect its bias to lie between the results from the NASA and DOMINO products, based on the magnitude of $NO_x$ emissions in 2010.

We evaluate the simulated ozone concentrations with global surface measurements from the TOAR database using three ozone metrics: maximum daily 8-hour average (MDA8) ozone, daytime average ozone ($8:00 - 20:00$ local time), and 24-hour average ozone. In addition to the GC-adj simulation, with which we derived top-down $NO_x$ emissions, we also input the same top-down emissions to GCv12 and evaluate ozone simulations from this more recent version of the GEOS-Chem that includes updated halogen and isoprene chemistry.

All GC-adj simulations of 2-meter ozone concentrations have a high bias compared to the TOAR measurements in 2010. NMB and Normalized Mean Square Error (NMSE) are largest for 24-hour ozone concentrations. Simulations using posterior $NO_x$ emissions have slightly better agreement with the measurements from TOAR in 2010 (Fig. 3). In particular, simulations using the DOMINO posterior $NO_x$ emissions have the smallest NMB in all ozone metrics and the smallest NMSE in all metrics except for the North Hemisphere (NH) summertime MDA8 ozone. Simulations using the NASA posterior $NO_x$ emissions have the best spatial correlation when compared with measurements for all metrics except for the NH summer daytime ozone and annual MDA8 ozone, for which DOMINO posterior simulations have the largest correlation coefficient (Fig. S4).

In comparison, GCv12 simulations have a low bias in daytime ozone, but high bias in 24-hour average ozone, reflecting the potential underestimate of ozone loss at night. The impact of $NO_2$ assimilation on improving estimates of surface ozone simulations in GCv12 depends upon the ozone metric, as shown in the bottom left panel of Fig. 3. Simulations using the DOMINO posterior emissions have the smallest NMB for annual mean daytime ozone; simulations using bottom-up $NO_x$ emissions have the smallest NMB for annual mean MDA8 ozone; simulations using the NASA posterior emissions have the smallest NMB for annual mean 24-hour averaged ozone. These results suggest that the simulated diurnal variations of surface ozone concentrations may not be correct. The current constraints on $NO_x$ emissions use observations from OMI, which overpasses the same location approximately once per day. The diurnal variations of $NO_x$ emission are constrained to be those of the prior emissions. The daily $NO_2$ column densities from OMI are smaller compared to the diurnally varying ground-based retrievals [Herman et al., 2019]. Assimilating $NO_2$ observations from instruments overpassing at different time of the day [e.g., Boersma et al., 2008; Lin et al., 2010; Miyazaki et al., 2017] and using hourly constraints from the geostationary satellite data (e.g., Geo-stationary Environmental Monitoring Spectrometer (GEMS), Tropospheric Emissions: Monitoring of Pollution (TEMPO) [Zoogman et al., 2017] and Sentinel-4) have the potential to improve simulations of ozone diurnal variations and different ozone metrics, although the ratio of $NO_2$ column densities from satellites that overpass in the morning and afternoon are generally lower than the same ratio from surface measurements [Penn and Holloway, 2020]. Simulated MDA8 ozone values are mostly biased low in NH summer but biased high in annual mean concentrations, reflecting different seasonal

variations in simulated and measured ozone concentrations, which will be further discussed in Sect. 3.2. Evaluations with the CNEMC ozone measurements in China are in Sect. S2.

## 260  3.2 Seasonal variation

The seasonal variations of monthly $NO_x$ emissions are consistent between the prior and the NASA posterior emissions (Fig. 4). The DOMINO posterior emissions show different seasonal variations in several regions. In China, the prior and the NASA posterior $NO_x$ emissions show summer peaks, which are mainly caused by the increase of natural sources when temperatures are high and lightning occurs more often [Qu et al., 2017]. The DOMINO posterior emissions have the largest values in January
and June in China, consistent with the posterior seasonality from Miyazaki et al. [2017] constrained by the same OMI $NO_2$ product. The June peak in China has been explained by the crop residual burning [Stavrakou et al., 2016]. The peak of the DOMINO posterior $NO_x$ emissions in the United States and Mexico shifted earlier in the year to June and July compared to the prior and NASA posterior emissions, similar to the results from Miyazaki et al. [2017]. The peak in DOMINO posterior emissions corresponds to the time of high soil $NO_x$ emissions, which are reported to be underestimated in high-temperature
agricultural systems in the bottom-up inventory [Oikawa et al., 2015; Miyazaki et al., 2017]. The differences between the DOMINO posterior and the other two sets of emissions are especially large during the springtime in India, when biomass burning activity increases [Miyazaki et al., 2017; Venkataraman et al., 2006]. These retrieval products have similar number of observations and spatial distributions of observation densities after the filtering. The different seasonal variations in the posterior $NO_x$ emissions may reflect the AMF structural uncertainties when the retrieved $NO_2$ column densities use different
ancillary data [Lorente et al., 2017]. For instance, the GEOS-Chem $NO_2$ SCDs converted using the scattering weight from the NASA product have larger seasonal variations than the SCDs converted using the DOMINO scattering weight in the US, reflecting the different seasonal variations of vertical sensitivities from the two retrievals. The seasonal variations of simulated surface $NO_2$ concentrations are similar with measurements in China and the US (see Fig. S6).

Seasonal variations of 2-meter ozone concentrations simulated by the GC-adj are also similar despite different $NO_x$ emission inputs: the differences in correlation coefficients of the simulated and the measured monthly ozone concentrations are less than 9%. The simulations of 2-meter ozone concentrations from GCv12 show better seasonality when using the posterior $NO_x$ emissions than using the prior, as shown in Fig. 5. Simulations using the CEDS inventory show double maxima in April and August, whereas surface measurements only show a single maximum in April. Assimilation of NASA $NO_2$ concentrations
alleviates this difference and leads to the largest correlation with measured MDA8 and 24-hour average ozone; simulations using the DOMINO posterior emissions have the largest correlation coefficient for daytime ozone. That being said, the correlation coefficients are not notably different. The August ozone peak in the prior simulation is mainly due to the high ozone concentrations in North China, Southwest China, and North India. The NASA and DOMINO posterior simulations have both reduced surface ozone concentrations in North China Plain and Northeast China in August due to the larger posterior $NO_x$
emissions than the prior in these high-$NO_x$ regions. Both posterior ozone simulations are also smaller than the prior in Tibet

and North India due to the reductions of posterior $NO_x$ emissions in low-$NO_x$ region. The August ozone peak in the DOMINO posterior comes from the higher ozone concentrations in Angola and Democratic Republic of the Congo compared to the NASA posterior and prior simulations in the same month and DOMINO posterior simulations in the previous months. This can be explained by the larger upward adjustment of DOMINO posterior $NO_x$ emissions in South Africa in August. These results show the large spatial heterogeneities on the responses of ozone seasonality to the changes in $NO_2$ abundances on a global scale. Compared with CNEMC measurements in China, simulations using the prior emissions have the most consistent seasonal variations and smallest NMSE. All simulations have smaller seasonal variations than the measurements in daytime ozone.

## 3.3 Inter-annual variations

The three different versions of $NO_x$ emissions have different regional trends from 2005 to 2016 as shown in Fig. 6. In China, the NASA posterior $NO_x$ emissions increased by 32% and the DOMINO posterior $NO_x$ emissions increased by 32% from 2005 to 2011. From 2011 to 2016, they decreased by 20% (NASA) and 11% (DOMINO). This turning point reflects the regulation of $NO_x$ emissions in China since the "11th 5-year plan" in 2011. In India, both posterior $NO_x$ emissions showed continuous increases (by 24% from the NASA posterior and 34% from the DOMINO posterior) from 2005 to 2016. The sources of $NO_x$ emissions in India are mainly from thermal power and transportation and are expected to continue increasing in the near future under current regulations [Venkataraman et al., 2018]. In the US, $NO_x$ emissions decreased by 24% (NASA) and 19% (DOMINO) from 2005 to 2010 and then flattened from 2010 to 2016. This slowdown in the total top-down $NO_x$ emissions was attributed to the growing contribution from industrial, areal, and off-road mobile sources as well as the slower than expected decreases in on-road diesel $NO_x$ emissions by Jiang et al. [2018]. Silvern et al. [2019], however, argued that the slowdown was driven by the weaker decreases in background sources of $NO_x$, which has increasing contribution with the decrease of anthropogenic $NO_x$ sources. In Mexico, the two posterior $NO_x$ emissions consistently increased by 6% (NASA) and 13% (DOMINO) from 2005 to 2016. The DOMINO posterior shows more obvious increase in Mexico from 2010 to 2016. This increase in Mexico is not reflected in the bottom-up estimates from the EPA National Emissions Inventory. In Australia, the NASA posterior increases by 10% from 2005 to 2016. In comparison, the DOMINO posterior decreases from 2005 to 2010 and increases afterwards, consistent with the posterior trend from Miyazaki et al. [2017]. The different trends in posterior $NO_x$ emissions are propagated from the trends in the two OMI $NO_2$ retrieval products. The discrepancies are likely due to the different surface albedo and cloud products used in the two retrievals, which affect averaging kernel sensitivities. The trends of $NO_x$ emissions in South America are different in the two posterior estimates after 2012, when the NASA posterior emissions started to decrease by 27% and the DOMINO posterior emissions started to increase by 11% up until 2016. In Western Europe and Africa, posterior $NO_x$ emissions fluctuate and do not have a significant consistent trend from the two inversions.

The magnitudes of DOMINO posterior $NO_x$ emissions are consistently larger than the NASA ones throughout the period. The 12-year averages of annual $NO_x$ budgets from NASA posteriors are 37% (China), 53% (India), 43% (US), 50% (Mexico), 45% (Australia), 58% (South America), 47% (Western Europe), and 46% (Africa) smaller than the DOMINO posterior.

We evaluate the trend of simulated surface $NO_2$ concentrations in the US with AQS measurements due to its availability throughout the study period (Fig. 7). From 2006 to 2016, the surface $NO_2$ concentrations show consistent decreases in the AQS measurements (by 32%) and GC-adj simulations (by 26% using the NASA posterior, by 10% using the DOMINO posterior, and by 7% using the prior emissions). Since we use the same anthropogenic emissions throughout 2006-2016 in the prior simulations, the variations in the black line reflect changes from natural sources and the impact of meteorological factors (e.g., temperature, humidity, wind, etc.). Surface $NO_2$ simulations using the NASA posterior $NO_x$ emissions also have the largest correlation coefficient when compared to the measurements ($R^2 = 0.93$ for the NASA posterior, $R^2 = 0.81$ for the DOMINO posterior, and $R^2 = 0.74$ for the prior). The more consistent trends and correlations in surface $NO_2$ simulations using the NASA posterior emissions are consistent with the larger decrease of NASA posterior $NO_x$ emissions in the US (by 20%, or for comparison a decrease of 1% in the DOMINO posterior) from 2006 to 2016, as shown in Fig. 6.

The interannual variability of global simulations of 2-meter ozone sampled at the TOAR locations is similar between GC-adj and GCv12. During the NH summer, simulations using the DOMINO posterior $NO_x$ emissions have the most consistent trend in daytime and 24-hour average ozone in both models (see Table S1); GC-adj simulations using the NASA posterior emissions have the best consistency with the measured trend of MDA8 ozone. The different performance of $NO_x$ emission datasets for different ozone metrics is a consequence of the hard constraint on $NO_2$ diurnal variations within the assimilation (and the lack of sufficient observations to constrain this). This can lead to better agreement of mean ozone concentration with measurements over particular hours but worse mean concentrations averaged over other hours. Detailed analyses of global ozone trends are in Sect. S3. At the regional scale, shown in Fig. 8, surface ozone measurements from TOAR mostly fall within the ranges of assimilation results. The interannual variations of simulated ozone over the whole region (black dotted lines) are generally smaller than the ones at grid cells that include surface measurements (black solid lines). The number of years that ozone measurements are available in each grid cell is shown in Fig. S8. The overlap of solid black and green lines in Fig. 8 suggests that interannual variations of anthropogenic $NO_x$ emissions from CEDS do not have a large impact on surface ozone simulations. The trends of simulated annual MDA8 ozone concentrations are correlated with impacts from meteorology and non-$NO_x$ sources based on simulations (shown as green lines) that use the same anthropogenic $NO_x$ emissions for all years and simulations that use interannually varied anthropogenic $NO_x$ emissions, leading to ozone changes of up to 4 ppbv (China), 5 ppbv (South Korea), 1ppbv (US), 2 ppbv (Mexico), 1 ppbv (South America), 1 ppbv (Australia), 1 ppbv (Western Europe), and 6 ppbv (Africa) from one year to the next. The trends of simulated MDA8 ozone are similar when using the NASA and the DOMINO posterior $NO_x$ emissions as inputs. The DOMINO-derived MDA8 ozone concentrations are higher than the NASA-derived ones in all studied regions, represented by the upper and lower limit of the error bars respectively. GCv12

simulated ozone concentrations are smaller than simulations from GC-adj, especially over relatively less polluted regions, consistent with the inclusion of halogen chemistry in GCv12, which depleted ozone. The simulated MDA8 ozone trends in grid cells that include measurements in the US and Australia are more consistent with the TOAR measurements than the other regions, with coefficients of determination ($R^2$) larger than 0.45. The larger differences in ozone between the prior and posterior emissions as well as variability between the two top-down $NO_x$ emissions in GCv12 suggest a larger responsiveness of the ozone chemistry to changes in $NO_x$. We do not expect simulated ozone trends to be completely consistent with the measurements in the TOAR database due to errors in the model's transport, chemical mechanism, and VOC emissions.

We further separate the ozone trends in grid cells that include measurements into changes caused by $NO_x$ emissions as well as meteorology and non-$NO_x$ sources. The second trend is calculated through simulations that use constant $NO_x$ emissions throughout the studied years. It has similar trend from GCv12 and GC-adj as shown in the green lines in Fig. 9. The trend caused by $NO_x$ emissions is obtained by subtracting the second trend from the ozone trend simulated using $NO_x$ emissions at each corresponding year. The ozone trends due to changes in meteorology and non-$NO_x$ sources (green lines) are moderately correlated ($R > 0.5$) with measurements from TOAR in Australia, the US, South America, and India. The ozone trends due to changes in posterior $NO_x$ emissions (red and blue lines) only have positive correlations with TOAR measurements in both GC-adj and GCv12 simulations in Africa and Australia. Ozone measurements in 2014 decreased compared to the 2006 level in the US and Mexico. GC-adj simulations do not have big trends in these regions, whereas GC-v12 simulations show increases in China, the US, and Mexico. Meteorological and non-$NO_x$ sources lead to larger inter-annual variations in ozone simulations than those driven by $NO_x$ emissions in South America, Australia, and Africa, where anthropogenic activities are much less than the other regions. These underscore the challenges of attributing observed variations in ozone to changes in $NO_x$ emissions at regional scales.

## 4 Western US remote ozone

Assimilations of ozone precursor gases have the potential to improve remote ozone simulations, which can be used to provide boundary conditions for regional air quality models and to quantify and attribute sources of background ozone. We therefore focus specifically on remote regions in the US in this section to evaluate the vertical profile and surface concentrations of ozone simulations.

### 4.1 Evaluations with ozonesonde profiles

Field campaigns and routine observations of ozone concentrations along the west coast of the US have provided opportunities to understand regional and intercontinental influences on surface air quality [Cooper et al., 2015]. Evaluations with the IONS-2010 measurements in Fig. 10 show that the GCv12 simulations of ozone vertical profiles have negative biases (NMB between -8% and -32%) above all six sites. The standard deviations of ozonesonde and simulated profiles overlap with each other (see

Fig. S9). The GC-adj simulations have positive biases at San Nicolas and Trinidad Head and have smaller negative biases (NMB between -3% and -11%) at the remaining sites than the GCv12 simulations. The magnitudes of the NMSE and NMB of the GCv12 simulations at 700 – 900 hPa are also larger than those of the GC-adj simulations (see Fig. S10). The prior simulations in GCv12 applies $NO_x$ emissions at different altitude, whereas the posterior GCv12 and all GC-adj simulations apply all $NO_x$ emissions to the surface. This leads to different transport and formation of ozone at different model layers and therefore causes larger differences in ozone simulations in the upper troposphere. The air masses at this altitude in the eastern Pacific are demonstrated to impact inland near surface ozone concentrations [Cooper et al., 2011; Lin et al., 2012; Yates et al., 2015]. The different biases in ozone simulations close to surface can be explained by the usage of different emission inventories (e.g., different biogenic emissions) and different boundary layer mixing scheme (non-local mixing [Lin and McElroy, 2010] in GCv12 and full mixing in GCadj). The different chemical mechanisms in the two model versions affect the different model biases especially in the upper troposphere. For instance, inclusion of halogen chemistry and additional chlorine chemistry in GCv12 leads to 19% and 7% decreases of global tropospheric ozone burden [Sherwen et al., 2016a; Wang et al., 2019]. GCv12 simulations using the CEDS emissions have smaller NMSE and NMB than the simulations using the posterior $NO_x$ emissions in all 6 sites in 2010. In comparison, the GC-adj simulations using the DOMINO posterior $NO_x$ emissions have the smallest NMSE and NMB at all sites except for San Nicolas and Trinidad Head, where the prior simulations have the smallest error and bias. Further evaluations with ozonesondes at Trinidad Head in 2016 are shown in Sect. S4.

## 4.2 Evaluations with TOAR surface ozone measurements at remote sites

To further evaluate the model performance under different geographical scenarios, we compare surface ozone simulations from GC-adj and GCv12 with observations from simple to complex environments. These include 1) Mauna Loa Observatory and Mt Bachelor Observatory at night, which represent the lower free troposphere; 2) Mt. Bachelor Observatory, Lassen Volcanic National Park, Great Basin National Park, and Sequoia / Kings Canyon National Park at daytime, representing high elevation rural sites during well-mixed daytime conditions. The coefficients of determination (see Table S2) between the simulations and the measurements are larger than 0.6 for all daytime ozone comparisons except for Mt. Bachelor Observatory. The correlation coefficients are smaller than 0.5 for all nighttime comparisons, reflecting the need to further improve simulations of nighttime chemistry and atmospheric processes.

In Fig. 11, the surface ozone concentrations from both GC-adj and GCv12 simulations have low biases compared to the surface measurements at remote sites. These low biases in the GCv12 simulations are consistent with its performances when evaluated with ozonesondes from IONS-2010 and with daytime surface ozone at the global scale. However, the low biases in the GC-adj simulations are different from its high biases when compared with the global surface ozone concentrations and the ozone profiles at San Nicolas and Trinidad Head. This demonstrates the different biases in ozone simulations at rural and urban sites. Simulations using the DOMINO posterior emissions have the smallest NMSE and NMB at all remote sites except for the GCv12 simulations at Mauna Loa at night and Great Basin during the daytime.

## 5 Discussion and conclusions

We performed global inversions of $NO_x$ emissions from 2005 to 2016 using two widely used OMI $NO_2$ retrievals from NASA (OMNO2 v3) and KNMI (DOMINO v2). Different vertical sensitivities from the two retrievals are a major cause of the discrepancies in the posterior emissions. The DOMINO posterior $NO_x$ emissions have larger magnitude than the prior and the NASA posterior. Consequently, GC-adj simulations using the DOMINO posterior $NO_x$ emissions have the smallest negative bias in surface $NO_2$ and the smallest positive bias in 2-meter ozone. The impact of $NO_2$ assimilations on improving estimates of the GCv12 surface ozone simulations depends upon the ozone metrics, suggesting inaccurate diurnal variations in the surface ozone simulations. GEOS-Chem simulations using the DOMINO posterior emissions have the largest coefficients of determination for summertime daytime ($R^2$=0.81) and summertime 24-hour ($R^2$=0.96) ozone. Simulations using the NASA posterior emissions have the smallest bias and error for all ozone metrics and the largest correlation for summertime MDA8 ozone ($R^2 = 0.88$). Ozone simulations with GEOS-Chem v12.1.1 using the DOMINO posterior $NO_x$ emissions lead to the most consistent seasonality in 24-hour average ozone ($R^2 = 0.99$) with TOAR measurements, while the NASA posterior emissions lead to the best agreement in seasonal variations of MDA8 ($R = 0.96$) and daytime ozone ($R = 0.98$). The interannual variations of posterior $NO_x$ emissions from the two products are similar in China, India, the US, Mexico and Australia, but different in South America, West Europe and Africa. Surface $NO_2$ simulations using the NASA posterior have the best agreement with measurements in the US. Daytime and 24-hour average ozone simulations using the DOMINO posterior also have the best trend ($R = 0.72$ and 0.88) in the Northern Hemisphere summer. The GC-adj simulations using the NASA posterior $NO_x$ emissions have the best trend in MDA8 ozone in NH summer.

Posterior $NO_x$ emissions lead to improved simulations of ozone at several remote sites in the western US. The GC-adj simulations using the DOMINO posterior emissions have the smallest NMSE and NMB compared to ozonesonde measurements during IONS-2010, except for the San Nicolas and Trinidad Head sites. At the remote surface sites evaluated in this study, surface ozone simulations using the DOMINO posterior emissions have the best performance except for GCv12 simulations at Mauna Loa at night and Great Basin during the daytime. The reduced negative biases in daytime surface ozone simulations using the DOMINO posterior emissions at these remote sites and at most IONS-2010 sites are consistent with the increases of daytime remote ozone in the western US through $NO_2$ and ozone data assimilation in Huang et al. [2015]. Simulations using the DOMINO posterior emissions are demonstrated to provide more precise magnitudes at these remote sites and can potentially be used as boundary conditions for regional air quality models for further air pollution and health studies.

The remaining differences between simulated and measured ozone can be explained by the roles of VOCs, errors in satellite retrievals, and uncertainties in the chemical and physical processes in the model simulations. In addition to $NO_x$, emissions of other ozone precursors also impact the accuracy of ozone simulations. For instance, inversion of isoprene emissions over the

southeast US decreases surface ozone simulations by 1-3 ppbv [Kaiser et al., 2018]. Inversion of non-methane VOC emissions changes surface afternoon ozone simulations by up to 10 ppbv in China [Cao et al., 2018]. Assimilation of multiple species (e.g, ozone, CO, $HNO_3$ and $SO_2$) together with $NO_2$ may improve posterior ozone simulations, but the performance of posterior simulations may depend on the chemical transport model, as shown in Miyazaki et al. [2020], where the GEOS-Chem adjoint model v35 shows mixed performance in correcting the bias between ozonesonde and posterior simulations between 850-500 hPa at different latitude band. Both OMI $NO_2$ retrievals employed in this study use $NO_2$ vertical shape factors from coarse resolution simulations, and therefore are biased low compared to in-situ measurements [Goldberg et al., 2017]. These retrievals also have not explicitly accounted for the aerosol optical effects, which are demonstrated to degrade the accuracy of $NO_2$ column concentrations when AOD is very high [Chimot et al., 2016; Liu et al., 2019; Cooper et al., 2019]. The differences in the magnitude of ozone concentrations from GC-adj and GCv12 reflect the impact of other species emissions and chemical mechanisms on the bias of ozone simulations. Previous studies also show that global simulations at coarse resolution are not able to capture the observed persistence of chemical plumes in the free troposphere on intercontinental scales, therefore leading to underestimates of remote ozone concentrations [Hudman et al., 2004; Zhuang et al., 2018].

Although biases, errors, seasonalities and inter-annual variations of ozone simulations have been improved in several cases through constraints on $NO_x$ emissions, there are still large discrepancies in the vertical profile and diurnal variations between ozone simulations and measurements. For instance, the different performances of each set of $NO_x$ emissions on the simulations of different ozone metrics reflect errors in the ozone diurnal simulations. The differences in ozone vertical profiles suggest errors in vertical transport in the model. These discrepancies could not be improved by adjusting only surface $NO_x$ emissions using observations at one time of the day, as performed in this study. Future geostationary satellite observations will provide opportunities to update $NO_x$ emissions at every hour. Separately constraining $NO_x$ emissions from surface (e.g., anthropogenic sources) and upper atmosphere (e.g., lightning sources [Pickering et al., 2016]) and implementing these posterior $NO_x$ emissions at their corresponding vertical levels can potentially improve the vertical profile of ozone simulations.

**Data Availability**

The OMI NO$_2$ NASA standard product is downloaded from GES DISC (https://atrain.gesdisc.eosdis.nasa.gov/data/OMI/OMNO2_CPR.003/). The DOMINO and QA4ECV NO$_2$ retrievals are from KNMI (http://www.temis.nl/airpollution/no2col/no2regioomi_v2.php, http://www.temis.nl/airpollution/no2col/no2regioomi_qa.php). Ozonesonde profiles from Shasta, Big Sur, Point Reyes, Joshua Tree and San Nicolas Island are available from the NOAA Global Monitoring Laboratory

(ftp://aftp.cmdl.noaa.gov/data/ozwv/Ozonesonde/2_Field%20Projects/CALNEX/

Ozonesondes from Trinidad Head are also available from the NOAA Global) Monitoring Laboratory( ftp://aftp.cmdl.noaa.gov/data/ozwv/Ozonesonde/Trinidad%20Head,%20California/100%20Meter%20Average%20Files/). Precompiled TOAR ozone data were downloaded from: https://doi.pangaea.de/10.1594/PANGAEA.876108.

**Author contribution**

Z.Q., D.K.H., O.R.C, and J.L.N. designed the research; Z. Q. performed the research and prepared the paper with help from all authors.

**Acknowledgements**

Z. Qu and D. K. Henze acknowledge funding support from National Aeronautics and Space Administration (NASA) HAQAST NNX16AQ26G and NASA ACMAP NNX17AF63G. Part of the computing resources supporting this work was provided by
500 the NASA High-End Computing (HEC) Program through the NASA Advanced Supercomputing (NAS) Division at Ames Research Center. Z. Qu would also like to acknowledge high-performance computing support from Cheyenne (doi:10.5065/D6RX99HX) provided by NCAR's Computational and Information Systems Laboratory, sponsored by the National Science Foundation.

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

**Table 1. Total NO$_x$ emission (anthropogenic + natural) budgets in 2010 [Tg N yr$^{-1}$]**

|  | Bottom-up | NASA posterior | DOMINO posterior | QA4ECV posterior |
|---|---|---|---|---|
| Global | 52.20 | 51.86 | 61.36 | 57.97 |
| China | 9.85 | 9.57 | 11.94 | 10.30 |
| US | 5.69 | 5.63 | 7.45 | 6.78 |
| India | 4.03 | 4.04 | 5.16 | 4.74 |
| Western Europe | 3.13 | 3.09 | 4.33 | 3.57 |


### Total bottom-up NO$_x$ emissions in 2010

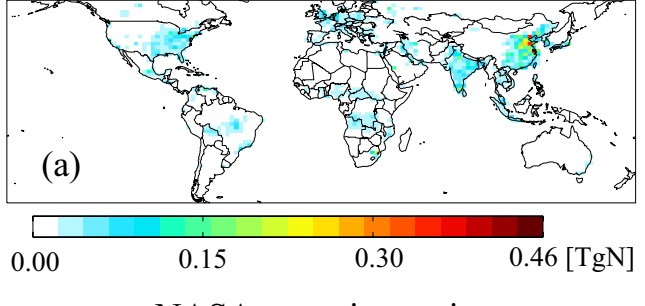

### NASA posterior - prior

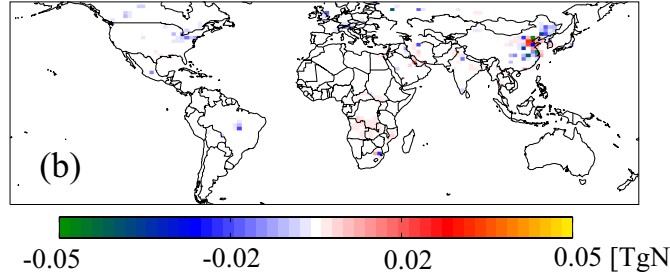

### DOMINO posterior - prior

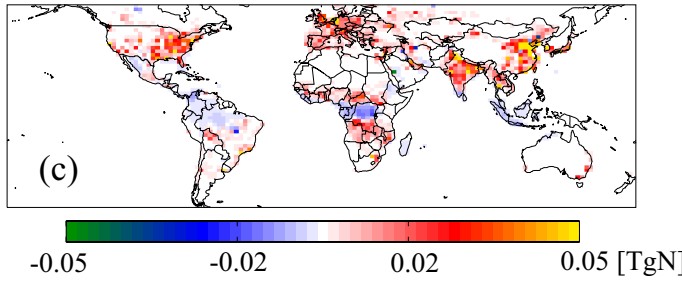

### QA4ECV posterior - prior

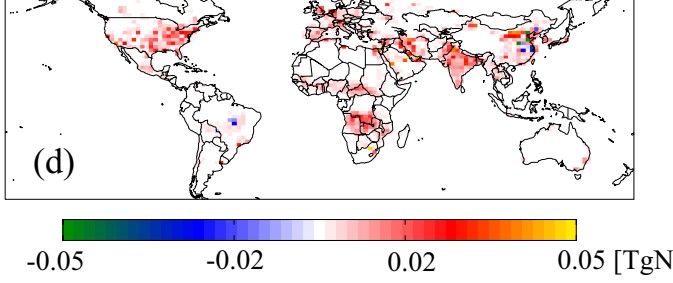

**Figure 1. (a) Global total NO$_x$ emissions from the bottom-up inventory and the differences between 4D-Var posterior and bottom-up estimates constrained by (b) NASA standard product v3, (c) DOMINO product v2, and (d) QA4ECV product in 2010.**


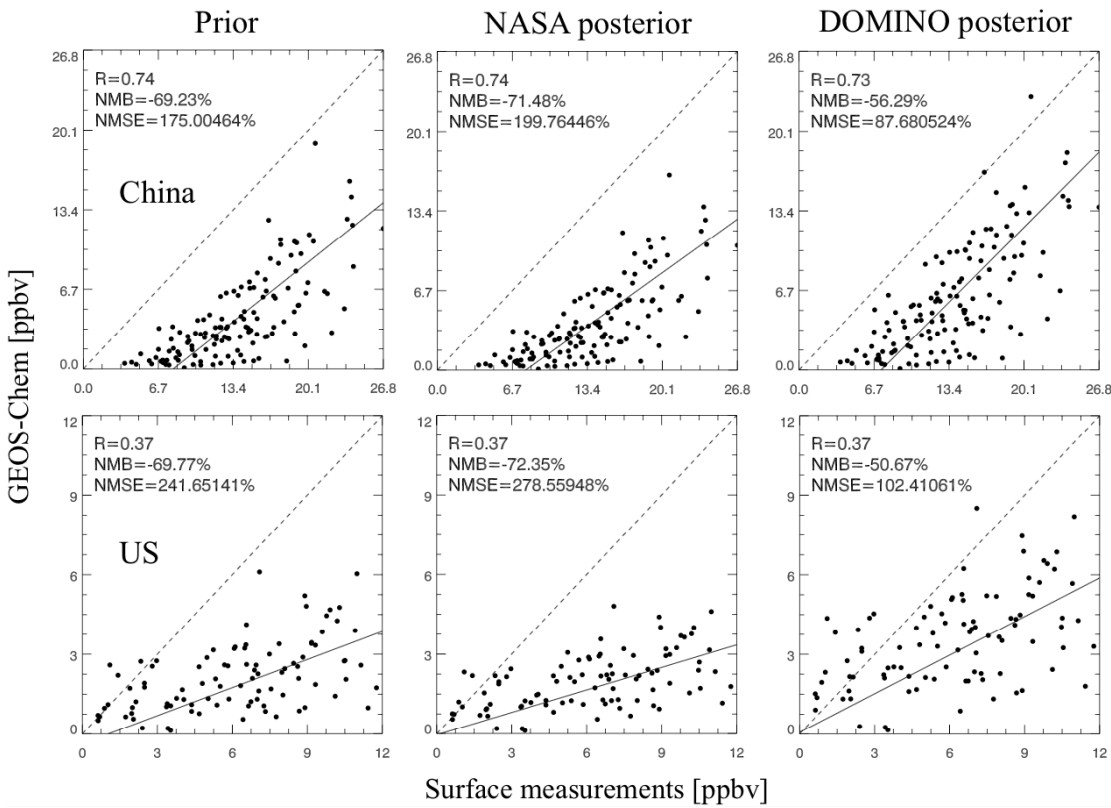

**Figure 2. Evaluation of annual mean surface NO₂ mixing ratios with measurements in China (top) and the US (bottom) in 2015.**

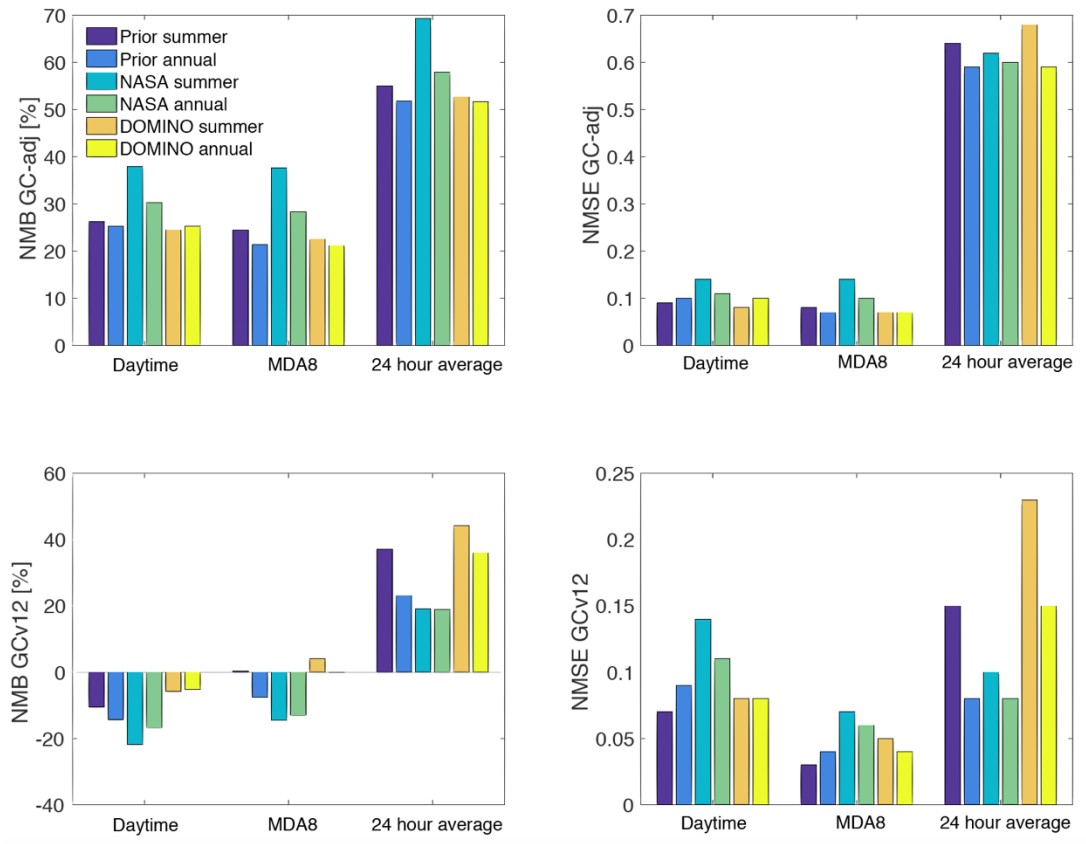


**Figure 3. NMB and NMSE of annual mean and NH summertime surface ozone concentrations when comparing all measurements from TOAR in 2010 with GC-adj (top) and GCv12 (bottom) simulations. The simulations are input with three sets of NO$_x$ emissions: CEDS bottom-up inventory (HTAP for GC-adj and CEDS for GCv12), posterior emissions constrained by the NASA product, and posterior emissions constrained by the DOMINO product.**


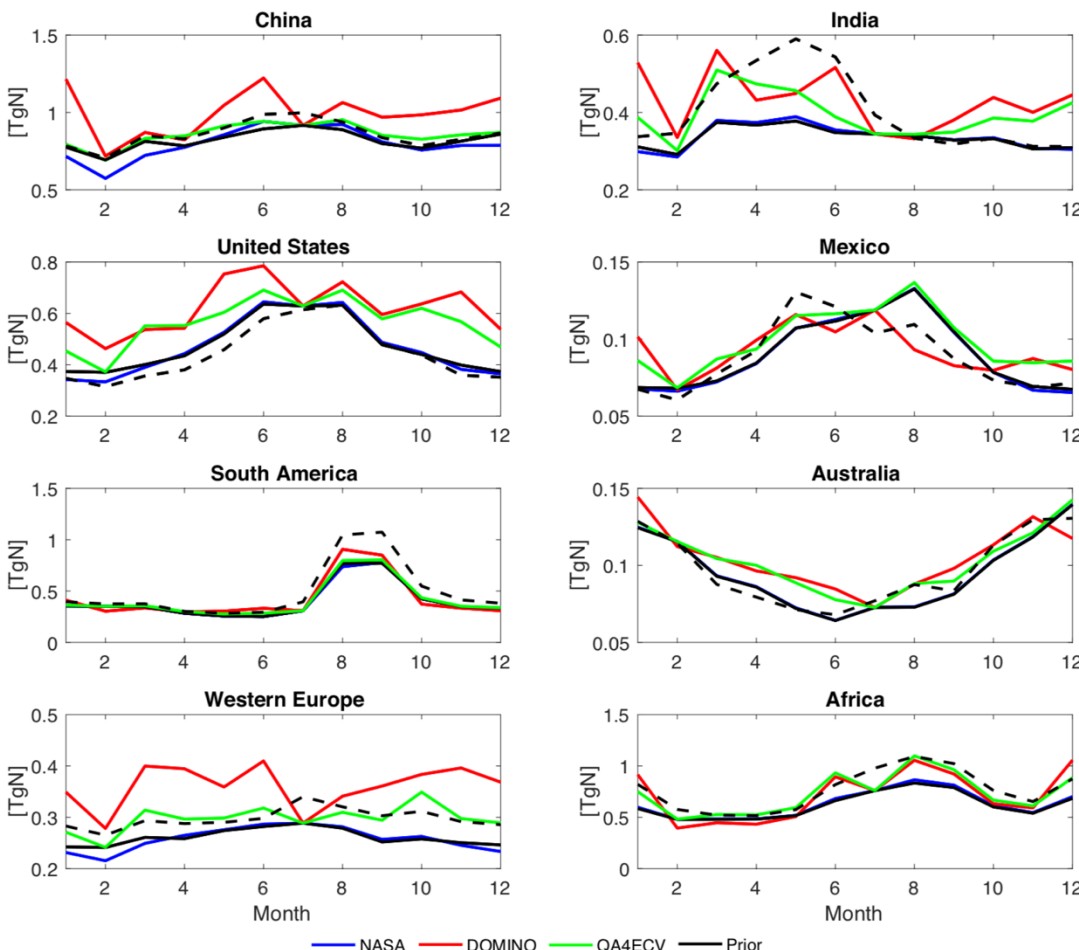

**Figure 4. Seasonal variations of total 4D-Var posterior NO$_x$ emissions in 2010. The black lines are prior emissions from bottom-up inventories (solid lines are from GC-adj, dashed lines are from GCv12). The blue lines are the emissions constrained by OMI NO$_2$ NASA product. The red lines are emissions constrained by OMI NO$_2$ DOMINO product. The green lines are emissions constrained by OMI NO$_2$ QA4ECV product.**

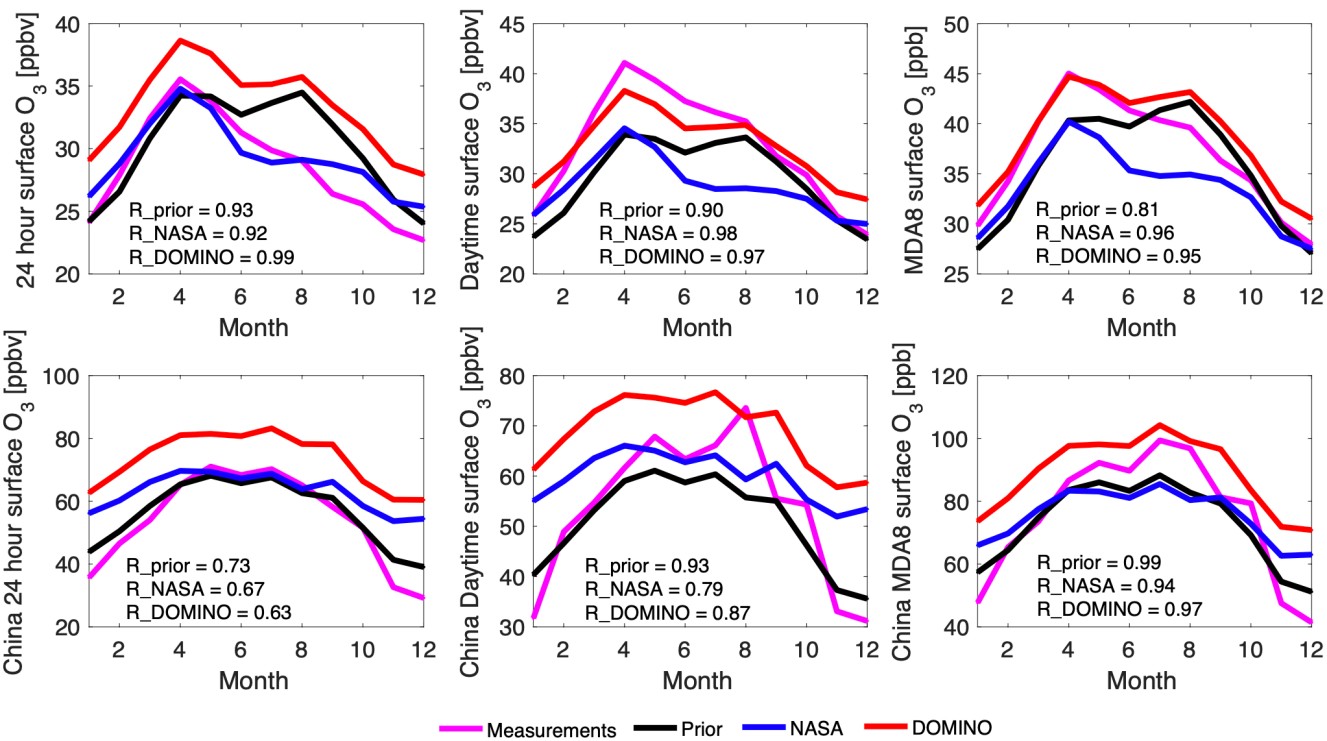


**Figure 5. Seasonality of surface ozone concentration at 2 meters in 2010 compared with TOAR (top) and in 2015 compared with CNEMC (bottom). Surface measurements are shown in magenta lines. Simulations are performed using GCv12 with NO$_x$ emissions from CEDS (black line), NASA posterior (blue line) and DOMINO posterior (red line).**

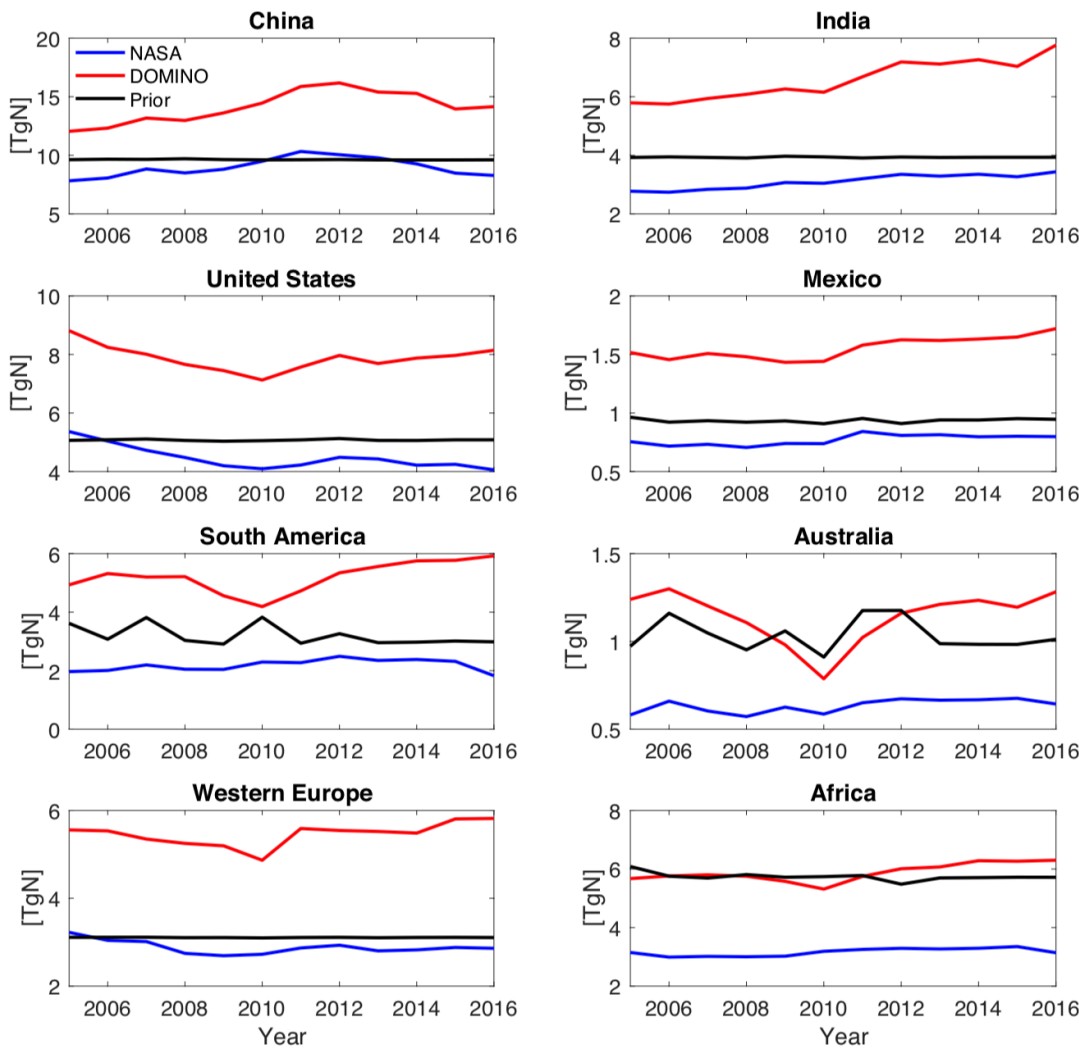


**Figure 6. Annual total posterior NO$_x$ emissions from 2005 to 2016. The black lines show prior total NO$_x$ emissions from bottom-up inventories, which use HTAP anthropogenic emissions in 2010 for all years. The blue lines represent the emissions constrained by the OMI NO$_2$ NASA product. The red lines represent emissions constrained by the OMI NO$_2$ DOMINO product.**


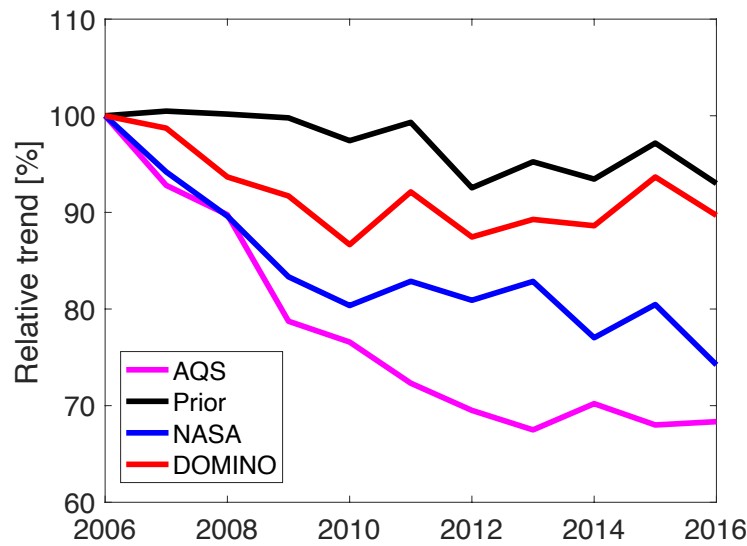

**Figure 7. The trend of annual mean surface NO₂ concentrations over the US from 2006 to 2016, expressed as a percent of the 2006 values. Surface measurements are from EPA AQS sites (magenta line). GEOS-Chem simulations are performed using prior emissions (black line) with constant anthropogenic emissions throughout the years, posterior $NO_x$ emissions constrained by NASA product (blue line), and posterior $NO_x$ emissions constrained by DOMINO product (red line).**


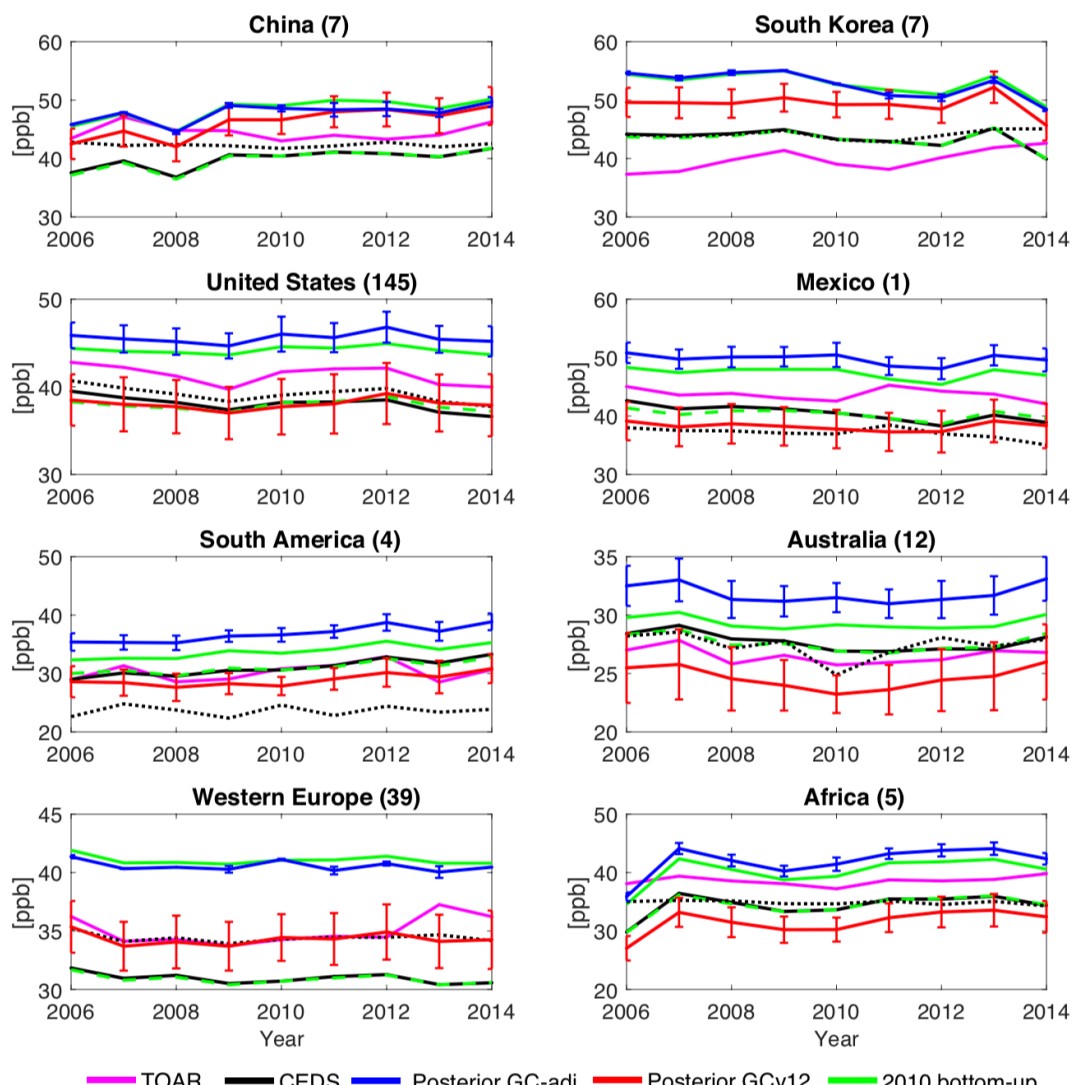

**Figure 8. The trends of regional mean annual MDA8 ozone concentrations from 2006 to 2014. Surface measurements are from the TOAR database (magenta line). Only sites that have continuous measurements throughout the 9 years are included. The numbers in the parenthesis are the number of 2° × 2.5° grid cells that include monitoring sites in each region. The black dotted lines show national mean of surface ozone from GCv12 simulations using the CEDS inventory. The other lines are simulations from GC-adj and GCv12 averaged over the 2° × 2.5° grid cells that include monitoring sites. Black lines show ozone simulations using the bottom-up $NO_x$ emissions from CEDS in each corresponding year. Green lines show ozone simulations using 2010 bottom-up $NO_x$ emissions for all years (HTAP 2010 for GC-adj shown in solid lines, CEDS 2010 for GCv12 shown in dashed lines). The vertical bars represent the spread of simulated surface ozone concentrations using the NASA and the DOMINO posterior $NO_x$ emissions.**

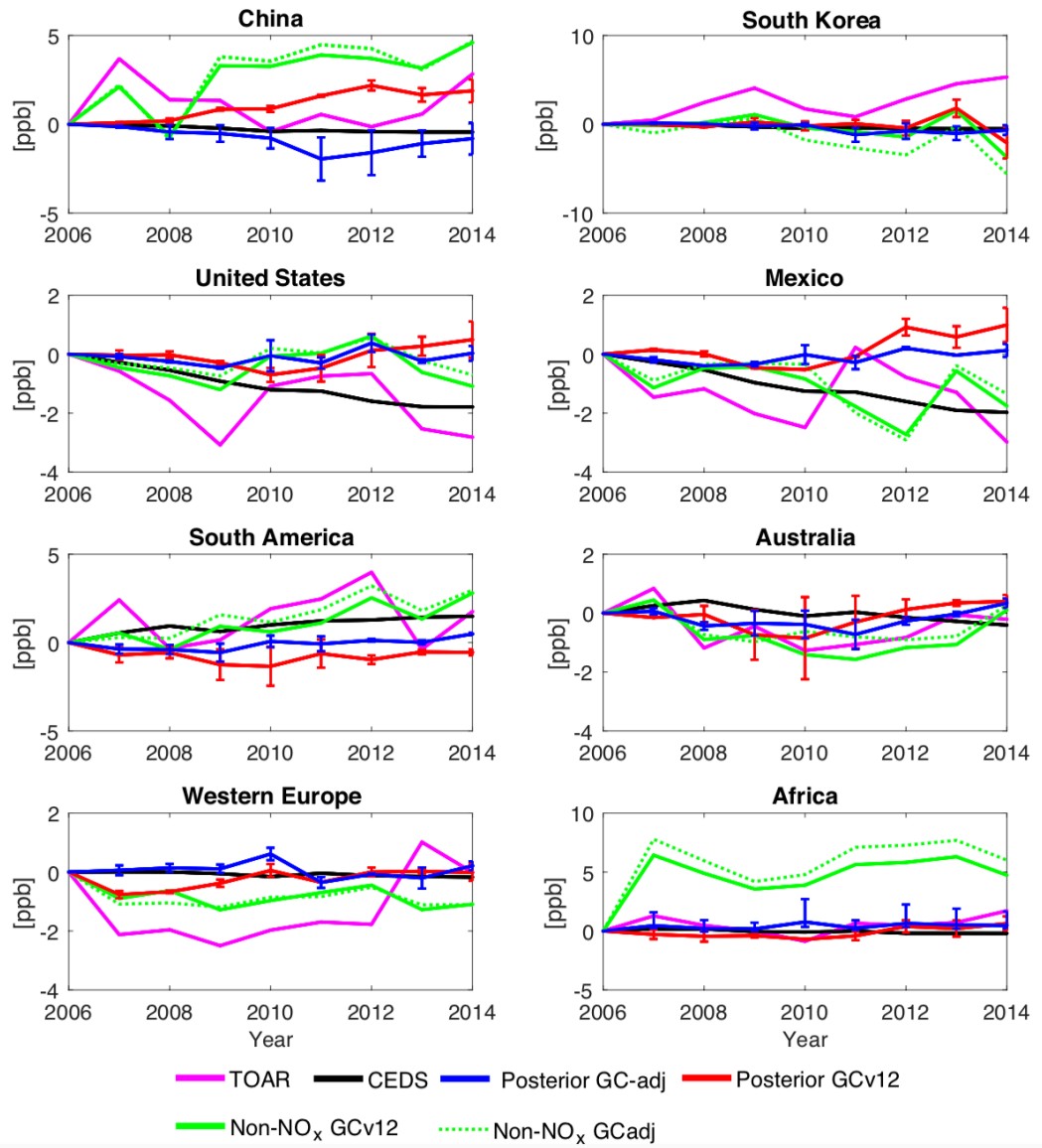

**Figure 9. Changes of regional mean annual MDA8 ozone concentrations compared to 2006 from TOAR measurements (magenta line), due to changes in bottom-up NO$_x$ emissions (black), due to changes in top-down NO$_x$ emissions (blue lines for simulations from GC-adj and red lines for simulations from GCv12), and due to changes in meteorology and non-NO$_x$ emissions (green lines). Only sites that have continuous measurements throughout the 9 years are included. The vertical bars represent the spread of changes from simulations using the NASA and the DOMINO posterior NO$_x$ emissions. The impact of meteorology and natural sources are removed from black, blue and red lines by subtracting simulations using 2010 bottom-up anthropogenic emissions for all years from simulations that use bottom-up NO$_x$ emissions corresponding to each year.**

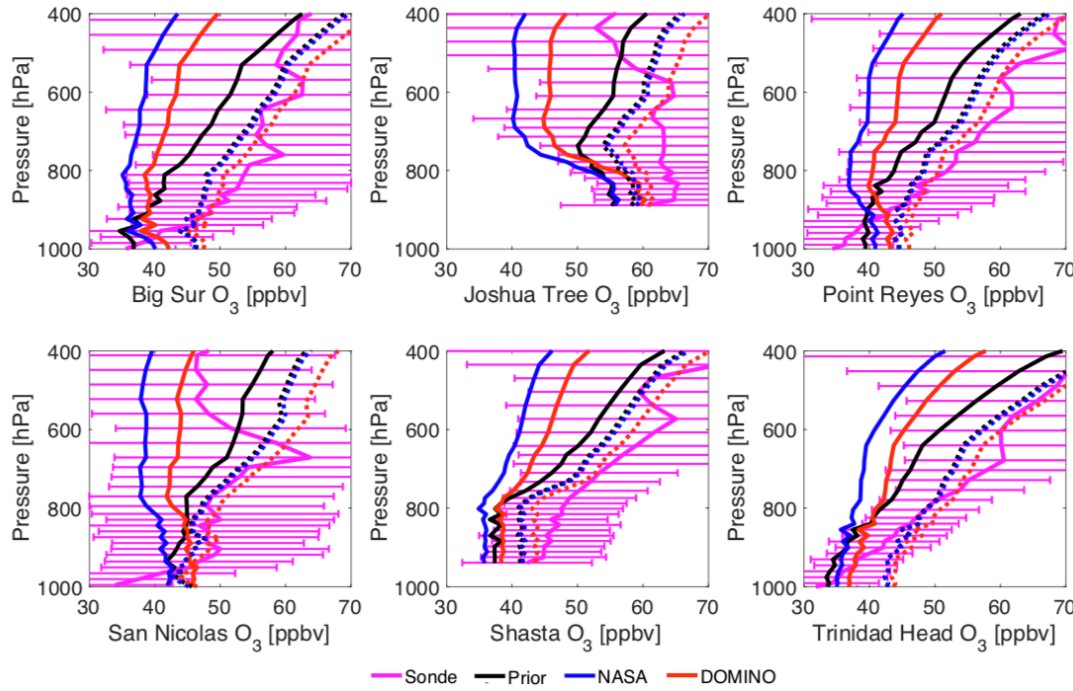


**Figure 10. Ozone vertical profiles averaged over May and June of 2010 from 6 ozonesonde measurement sites from the IONS-2010 field experiment in California. The six sites are over remote regions and are used to evaluate the intercontinental transport of ozone. Solid black (prior), blue (NASA posterior) and red (DOMINO posterior) lines are from the GCv12 simulations (prior anthropogenic**
**emission from CEDS), whereas dashed lines are from the GC-adj simulations (prior anthropogenic emission from HTAP). The horizontal bars show the standard deviations of the measurements at each vertical layer.**

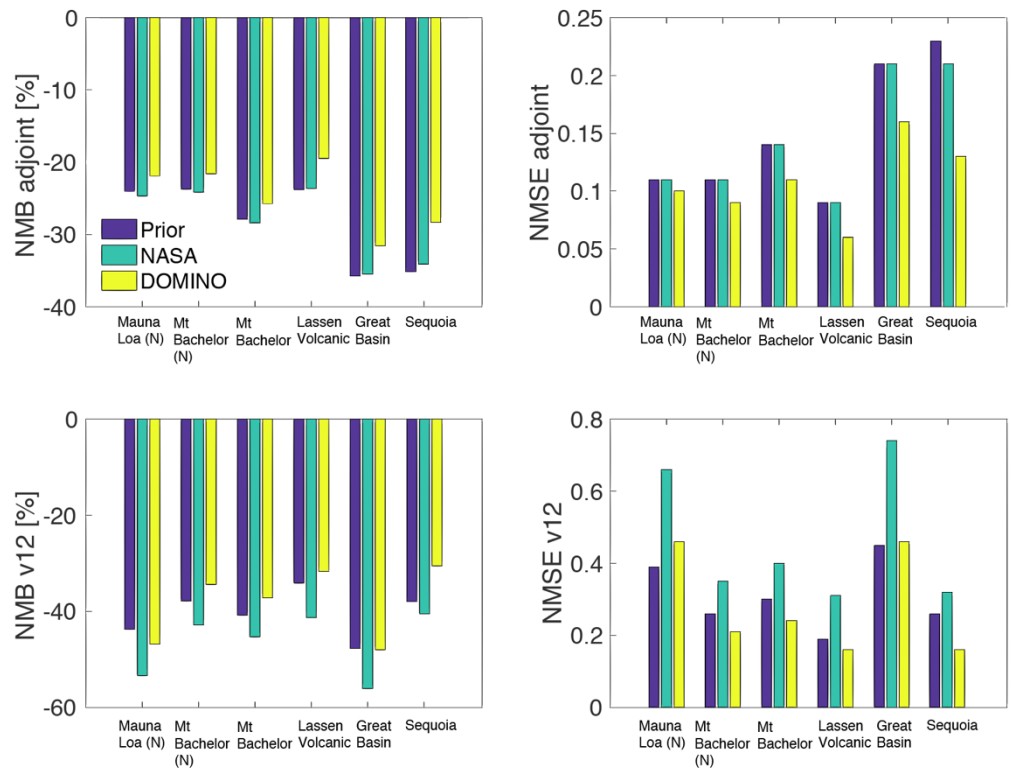

**Figure 11. NMSE and NMB of GC-adj (top) and GCv12 (bottom) ozone simulations in 2010 -2014 evaluated with surface measurements at remote sites. Three sets of NO$_x$ emissions, i.e., bottom-up inventory (HTAP for GC-adj, CEDS for GCv12), DOMINO posterior, and NASA posterior, are input in each model.**