# Peer review of "Impacts of global NOx inversions on NO2 and ozone simulations"

_Atmospheric Chemistry and Physics, 2020_

## Referee Comment (RC1) · Anonymous Referee #1 · 29 Apr 2020

Review of 'Improving NO2 and ozone simulations through global NOx emission inversions' by Zhen Qu et al.

**General comments**

Qu et al. have studied the impact of top-down NOx emission estimates derived from two OMI NO2 satellite data sets (NASA SP v3 and DOMINO v2) on NO2 and O3 simulations with the GEOS-Chem model. Previous work already showed (e.g. Verstraeten et al. [2015], studies by Miyazaki et al.) that O3 in the troposphere is generally better understood when NOx emissions are derived from satellite NO2 data than when taken from emission inventories.

Here, Qu et al. find substantial differences in the agreement of NO2 and O3 simulations

against independent measurements depending on whether data set NASA or data set DOMINO is used. This was to be expected given that it is well-known that the NASA and DOMINO datasets have considerable differences. A useful aspect of the study is that the authors now quantify the consequences of these differences, which is relevant because satellite data is increasingly used to improve model understanding of atmospheric composition.

What is disappointing however is that we do not learn much new. Simulations with the NASA emissions compare better to some metrics, and worse to others, but the authors do not explain why. This makes the manuscript a technical document, where it is left to the reader to figure out what emissions could work best for his/her particular purpose, without actual guidance on why that would be. The authors should do more to investigate why using one dataset leads to better agreement e.g. for surface O3 at remote sites, and the other for polluted sites. Aspects of spatial resolution, temporal representativeness, and vertical sensitivity should be taken into account when providing this guidance to potential users.

Another criticism is that the chain of technicalities is very long and that the experiments are set-up in a sub-optimal manner (for example comparing 2.5 degree simulations of surface NO2 to surface stations that are representative for much smaller domains). A major concern I have is with the lack of detail and clarity on how the adjoint incorporates the information from the satellite retrievals. From the manuscript I first suspected that monthly mean column NO2 data was simply used to estimate the emissions, suggesting that the highly variable and non-linear vertical sensitivities of the retrievals have not been used to interface the model with the satellite data. There are various studies pointing out how crucial it is to account for the vertical sensitivity of the NO2 retrievals, e.g. Miyazaki et al. [2017], Boersma et al. [2016] to name a few. Then I read the supplementary material and there the impression was given that at least the a priori profile shapes are made consistent between the NASA and DOMINO retrievals, but it remains unclear to what extent this has harmonized the data, and to what extent

vertical sensitivities between the two datasets are still fundamentally different.

**Specific comments**

P2, L42-43: the formation depends not only on the local NOx and VOC concentrations, but also on the radiative regime in which these occur.

P2, L65: different → differ

P3, L72: import →importer

P3, L78-81: Zhang et al. [2008] and Verstraeten et al. [2015] already showed that through optimizing NOx emissions in China, the simulated O3 over the Pacific and over the western US indeed improved.

*Section 2.1* It is unclear in this manuscript how the adjoint accounts for (a) vertical sensitivity of the satellite retrievals, and (b) the diurnal cycle of NOx emissions. These aspects are important enough to describe in the manuscript, for (a) useful information is provided in the supplement but it is not clear whether the replacing of the a priori profiles by GEOS-Chem prior profiles was also applied in the research to estimate the emissions. The authors should clarify this in section 2, and also briefly quantify to what extent the differences in prior simulations have been minimized by this approach. For (b), some info is given but only late in the game (P7: The diurnal variations of NOx emission are constrained to be those of the prior emissions), and we do not learn what the diurnal cycle is in the first place. Please revise section 2 thoroughly with this in mind.

Then I have other questions:

- how does the adjoint approach account for other relevant aspects of data assimilation?

- how is the OMI data averaged spatially to the grid of GEOS-Chem, and how are superobservation errors incorporated?

[Figure]

- did the authors only use the mostly cloud-free OMI retrievals?

Section 2.2: OMI is suffering from the so-called row anomaly, which was absent until mid-2007, and then became gradually more important. How did the authors ensure that the growing impact of the row anomaly did not unduly affect their trends in NOx emissions?

Section 2.3: it remains unclear what type of surface station was used for the GEOS-Chem surface evaluation. Using urban background and regional stations seems appropriate to evaluate the large GC grid cells, but urban street stations should be excluded.

P5, L152-154: what explains the OMI-driven differences between the posterior NOx emissions, differences in tropospheric slant columns or in the AMFs? Presumably the latter, but since the a priori profile differences have been "minimized", the differences must be in the assumptions on surface albedo and clouds. It would be best if the authors could shed more light on how the scattering weights or averaging kernels are different between the OMI NO2 retrievals. Please clarify.

P6, L173-174: the statement that "NO2 column simulations at $2° \times 2.5°$ in this study are likely to be underestimated and lead to high biases of posterior NOx emissions to match satellite NO2column concentrations" needs more evidence. The hypothesis that instant dilution leads to too much OH (by Valin et al. [2011]) may be valid for isolated NOx sources in otherwise pristine areas, but instant dilution of NOx emissions situated in high-background NO2 regions such as the eastern US or western Europe is probably of less concern.

P6, L193: what is the magnitude of the correction factors over China and the US? How do they vary by season?

P7, L195-199: this part is rather inconclusive. The GEOS-Chem simulations have been corrected for resolution (an increase) and surface measurements have been corrected down for molybdenum interference, and still GEOS-Chem with posterior emissions is

biased low by 20%-50%. What explains the persistent low bias?

P7, L224-225: OMI measurements frequently miss the high values of NO2 column densities that occur before or after its overpassing time. OMI was never designed to measure NO2 before or after its overpass time, so to say that OMI misses these high values is misleading. Please rephrase.

P7, L226: twice-per-day constraints on NOx emissions have been achieved in earlier studies based on SCIAMACHY + OMI (Boersma et al. [2008], GOME-2 + OMI [Lin et al., 2011], including via sophisticated assimilation schemes [Miyazaki et al., 2017].

P8, L237: the June peak in NO2 over China can be easily traced back to crop residu burning in that month – e.g. Stavrakou et al. [2016].

P8, L238-240: can you explain more why the DOMINO product would be more sensitive to soil NOx emissions? It's not because of the different a priori profiles assumed in the NASA and DOMINO retrievals?

P8, L243-244: please see my previous comment. The authors seem to know something very interesting here, but they don't show it. Is there any evidence that one retrieval would be more sensitive to NOx sources than the other? That would be extremely relevant to know more about. Since the satellite measurements are identical for the NASA and OMI retrievals, it must have to do with AMF differences , see e.g. Lorente et al. [2017]. But what drives the apparent difference in sensitivity – albedo, cloud fraction, cloud pressure?

P8, L246-256: Figure 5 – the daytime O3 simulations in China all seem strongly low biased relative to the observations. The other ozone metrics in China and all in the US match much better. Why is this?

P9, L271-272: "also not reflected"?

P9, L276: no reduction of NOx emissions in Europe? This is strange – NO2 tropopsheric columns are decreasing over Europe, and Miyazaki et al. [2017] showed reduc-

tions in for NOx emissions. Overall, Figure 6 looks very odd to me. DOMINO NO2 columns are 40% higher than NASA, but the NOx emissions inferred from DOMINO are more than 40% higher than the emissions inferred with NASA (L278-281). Also, Miyazaki et al. [2017] (Figure 9) still find reductions in NOx emissions over Europe between 2005 and 2014 based on the same DOMINO data, so how can you find increases? Please clarify.

P10, L295-297: I'm missing an explanation or hypothesis why NOx emissions from one dataset would do better than the other for different ozone metrics.

P10, L304 and L315: please clarify how the impact of meteorology and non-NOx sources on O3 changes was evaluated.

L306-307: "The trends of simulated MDA8 ozone are similar when using the NASA and the DOMINO posterior NOx emissions as inputs" – yes, but please also explain why the magnitude of the NASA-derived MDA8 O3 levels are biased high then.

P11, L332-333: the prior simulated O3 profiles in Figure 10 agree much better with the O3 sondes between 800-400 hPa than the assimilated profiles. I don't understand why that is, since the effect of the updated NOx emissions should be mostly felt in the lower 2 kms of the atmosphere. Or is this the impact of changes in background O3 in response to changing Asian emissions?

P13, L394-395: one important difference between this research and the work done by Miyazaki in a number of papers, is that the latter assimilates also other species relevant for NOx inversions and O3 simulations (e.g. CO, HNO3, SO2). It would be interesting to also discuss to what extent these additional constraints can help explain the "remaining differences between simulated and measured ozone".

P13, L398-400: the statement "Both OMI NO2 retrievals employed in this study use NO2 vertical shape factors from coarse resolution simulations, and therefore are biased low compared to in-situ measurements [Goldberg et al., 2017]." Brought up the

question (again) whether both OMI NO2 retrievals are at least consistent now in their use of the same coarse-resolution vertical shape factors (i.e. those from GEOS-Chem).

P13, L401: "retrievals also have not explicitly accounted for the aerosol optical effects, which are demonstrated to degrade the accuracy of NO2 column concentrations". This is an overstatement. Only when AOD is very high (>0.5-1.0) there are indications that implicit corrections break down. Even in Liu et al. [2019] accounting for AOD did not solve the low bias in tropospheric NO2 which was not apparent in the DOMINO scheme without an explicit aerosol correction.

**References**

Boersma, K. F., Vinken, G. C. M., and Eskes, H. J.: Representativeness errors in comparing chemistry transport and chemistry climate models with satellite UV–Vis tropospheric column retrievals, Geosci. Model Dev., 9, 875-898, doi:10.5194/gmd-9-875-2016, 2016.

Miyazaki, K., Eskes, H., Sudo, K., Boersma, K. F., Bowman, K., and Kanaya, Y.: Decadal changes in global surface NOx emissions from multi-constituent satellite data assimilation, Atmos. Chem. Phys., 17, 807–837, https://doi.org/10.5194/acp-17-807-2017, 2017.

Stavrakou, T., Müller, J.-F., Bauwens, M., De Smedt, I., Lerot, C., Van Roozendael, M., Coheur, P.-F., Clerbaux, C., Boersma, K. F., van der A., R. J., and Song, Y.: Substantial underestimation of post-harvest burning emissions in the North China Plain revealed by multi-species space observations, Sci. Rep., 6(32307), doi:10.1038/srep32307, 2016.

---

## Referee Comment (RC2) · Anonymous Referee #2 · 8 Jun 2020

This manuscript has presented top-down estimates of global NOx emissions using two OMI satellite NO2 products over 2005-2016 and using the GEOS-Chem adjoint inversion method. Considerable differences are found between the two top-down emission estimates. Implementing the top-down NOx emissions to the GEOS-Chem atmospheric chemistry model shows some improvements on the model simulation of tropospheric ozone. The study also points out that model improvements largely depend on the top-down emissions, the ozone metrics used, and model versions.

The manuscript is in general well organized and meets the scope of ACP. One main concern is that the manuscript has been presented as a model evaluation paper that comparing several model simulations with different NOx emissions with surface

and sonde ozone measurements. It lacks some analyses in depth to understand the driving factors of the differences. The key findings of this study are also not clear. Do we have a better understanding of the NOx emission trends as constrained by the satellite measurements, or how NOx emission changes affect tropospheric ozone? I think the concern and the following specific comments should be addressed before considering publish.

**Specific comments:**

1) Page 1, Line 24-25 in the Abstract:
The statement "using NOx emission datasets that have the best performance . . ." is not clear. As ozone simulation is affected by many other factors, the NOx emissions that have the best performance on ozone simulation may not be the correct one. Some results in this study also showed that satellite constrained NOx emissions did not necessarily improve ozone simulation (e.g., China daytime surface ozone in Figure 5)

2) Page 3, Section 2.1:
What was the spin-up time for the model simulations? Were you using the same initial conditions? Please clarify.

3) Page 6, Line 179:
Should here "the average of GEOS-Chem simulated NO2 column density" be OMI observed NO2 column density over 2x2.5 grid cell? Here you are generating pseudo measurements in the statement. The ratio should be calculated by OMI observations to avoid the OMI vs. model biases.

4) Page 8, Line 240-245:
The large differences in seasonal variations of DOMINO and NASA posterior NOx emissions seem interesting. Here you explained that the DOMINO posterior may

better constrain soil emissions. Do you have any evidence or support for that?

5) Page 8, Line 250-256:
Here you showed that prior simulated surface ozone concentrations had double maxima in April and August, and the posterior results partly corrected the biases. What cause the double maxima in the prior simulation? And how NOx emission changes correct the August maximum? Please clarify.

6) Page 9, Line 269-271:
As indicated in Figure 6, interannual changes in the two posterior NOx emissions in Australia over 2005-2016 are not that consistent. The DOMINO results show large reduction over 2006-2010 and then increase afterwards. Do you have any explanation why the two satellite products show different interannual variation and trends over some regions?

7) Page 10, Line 319:
"Ozone measurements in 2014 decreased compared to the 2006 level in China, the US, South America and Mexico". I do not see from Figure 9 that in China ozone concentration in 2014 was lower than 2006.

8) Page 10, Line 314-316: How did you separate the ozone trends caused by NOx emissions vs. meteorology? A description in the main text is needed. Also, you may calculate the meteorology (non-NOx) effects using either GC-adj or GCv12 results? Which one did you use in Figure 9, and how they differed?

9) Page 11, Line 338: It is surprising that the model versions (GCadj and GCv12) simulate very different ozone vertical profiles. GCv12, which is a more updated version, has much large biases in the upper troposphere, in particular with the updated NOx emissions. Can you explain why in GCv12 changes in surface NOx emissions

would lead to large ozone changes in the upper troposphere?

---

## Author Comment (AC1) · 8 Aug 2020

We have responded to each comment below. Our replies are in blue, and the revised manuscript text is written in bold.

**Response to review 1**

Qu et al. have studied the impact of top-down  $NO_x$  emission estimates derived from two OMI  $NO_2$  satellite data sets (NASA SP v3 and DOMINO v2) on  $NO_2$  and O3 simulations with the GEOS-Chem model. Previous work already showed (e.g. Verstraeten et al. [2015], studies by Miyazaki et al.) that O3 in the troposphere is generally better understood when  $NO_x$  emissions are derived from satellite  $NO_2$  data than when taken from emission inventories.

Here, Qu et al. find substantial differences in the agreement of  $NO_2$  and O3 simulations against independent measurements depending on whether data set NASA or data set DOMINO is used. This was to be expected given that it is well-known that the NASA and DOMINO datasets have considerable differences. A useful aspect of the study is that the authors now quantify the consequences of these differences, which is relevant because satellite data is increasingly used to improve model understanding of atmospheric composition.

What is disappointing however is that we do not learn much new. Simulations with the NASA emissions compare better to some metrics, and worse to others, but the authors do not explain why. This makes the manuscript a technical document, where it is left to the reader to figure out what emissions could work best for his/her particular purpose, without actual guidance on why that would be. The authors should do more to investigate why using one dataset leads to better agreement e.g. for surface O3 at remote sites, and the other for polluted sites. Aspects of spatial resolution, temporal representativeness, and vertical sensitivity should be taken into account when providing this guidance to potential users.

We appreciate the comments from the reviewer. We have modified the title, abstract, and the details in the manuscript accordingly to address these concerns. The title is now changed to "**Impacts of** global NOx inversions **on NO2 and ozone simulations**."

"Abstract. Tropospheric NO2 and ozone simulations have large uncertainties, but their biases, seasonality and trends can be improved with NO2 assimilations. We perform global top-down estimates of monthly NOx emissions using two OMI NO2 retrievals (NASAv3 and DOMINOv2) from 2005 to 2016 through a hybrid 4D-Var / mass balance inversion. Discrepancy in NO2 retrieval products is a major source of uncertainties in the top-down NOx emission estimates. The 12-year averages of regional NOx budgets from the NASA posterior emissions are 37% to 53% smaller than the DOMINO posterior. Consequently, the DOMINO posterior surface NO2 simulations greatly reduced the negative biases in China (by 15%) and the US (by 22%) compared to surface NO2 measurements. Posterior NOx emissions show consistent trend over China, US, India, and Mexico constrained by the two retrievals. Emission trends are less robust over South America, Australia, Western Europe and Africa, where the two retrievals show less consistency. NO2 trends have more consistent decreases (by 26%) with the measurements (by 32%) in the US from 2006 to 2016 when using the NASA posterior. The performance of posterior ozone simulations using NASA-based emissions alleviates the

double peak in the prior simulation of global ozone seasonality. The higher abundances of NO2 from the DOMINO posterior increase the global background ozone concentrations and therefore reduce the negative biases more than the NASA posterior in the GEOS-Chem v12 simulations at remote sites. Compared to surface ozone measurements, posterior simulations have more consistent magnitude and interannual variations than the prior estimates, but the performance from the NASA-based and DOMINO-based emissions varies across ozone metrics. The current hard-constraints on NOx diurnal variations and limited availability of remote sensing data hinder improvement of ozone diurnal variations from the assimilation, and therefore have mixed performance on improving different ozone metrics. Additional improvements in posterior NO2 and ozone simulations require more precise and consistent NO2 retrieval products, more accurate diurnal variations."

From a data user perspective, this work quantifies how differences in NO2 retrieval products propagate to the downstream estimates in top-down NOx emissions and ozone simulations. The discrepancy found in this study is larger than uncertainties caused by data assimilation methods (4D-Var versus Kalman Filter) and chemical transport models [Koukouli et al., 2020], and is therefore a unique contribution of this work. Detailed investigation of the origin of differences in the NASA and KNMI NO2 retrieval products goes beyond scope of this study. We do note however "**The GEOS-Chem NO2 SCDs converted using scattering weight from the NASA product are larger than the SCDs calculated using the DOMINO scattering weight and the same GEOS-Chem VCDs (See Fig. S2). These can be explained by the use of different surface albedo and cloud product in the two retrievals." (Added in Section 3)**

Another criticism is that the chain of technicalities is very long and that the experiments are setup in a sub-optimal manner (for example comparing 2.5 degree simulations of surface  $NO_2$  to surface stations that are representative for much smaller domains).

Comparing NO2 simulations at 2.5° with in-situ measurements is sub-optimal, but this is the highest resolution we can perform global 4D-Var assimilation using this model.

A major concern I have is with the lack of detail and clarity on how the adjoint incorporates the information from the satellite retrievals. From the manuscript I first suspected that monthly mean column NO2 data was simply used to estimate the emissions, suggesting that the highly variable and non-linear vertical sensitivities of the retrievals have not been used to interface the model with the satellite data. There are various studies pointing out how crucial it is to account for the vertical sensitivity of the NO2 retrievals, e.g. Miyazaki et al. [2017], Boersma et al. [2016] to name a few. Then I read the supplementary material and there the impression was given that at least the a priori profile shapes are made consistent between the NASA and DOMINO retrievals, but it remains unclear to what extent this has harmonized the data, and to what extent vertical sensitivities between the two datasets are still fundamentally different.

To clarify, we added the following sentences to Section 2.2:

"We converted GEOS-Chem NO2 VCD to SCD using scattering weight (NASA product) and averaging kernel (DOMINO and QA4ECV product) from the OMI retrievals and then compare GEOS-Chem SCD with SCD retrieved from OMI. A cost function is defined as the observation error weighted differences between simulated and retrieved NO2 SCD, plus the prior error weighted departure of the emission scaling factors from the prior estimates. We minimize the cost function using the quasi-Newton L-BFGS-B gradient-based optimization technique [Byrd et al., 1995; Zhu et al., 1994], in which the gradient of the cost function with respect to the control parameter is calculated using the adjoint method. Details of the assimilation of NO2 slant column densities (SCDs), how vertical sensitivities of satellite retrievals are accounted for, and the hybrid 4D-Var / mass balance inversion of NOx emissions are described in Qu et al. [2017]."

More detailed technicalities have been described in our previous publications cited in the manuscript and are therefore not the focus of this manuscript. The focus here is to apply this method for global  $NO_x$  inversion, evaluate the impact of different retrieval products on top-down emission estimates, and how the changes in  $NO_x$  emissions affect ozone simulations. Therefore, we did not repeat all the technical details that can be found in the cited publications. Please see our detailed responses below for all the concerns raised by the reviewer.

**Specific comments**

P2, L42-43: the formation depends not only on the local  $NO_x$  and VOC concentrations, but also on the radiative regime in which these occur.

Changed to "Ozone formation and trends depend nonlinearly on the local relative abundances of NOx and VOCs **and the radiative regime in which these occur**."

P2, L65: different  $\rightarrow$  differ

**Modified as suggested.**

P3, L72: import  $\rightarrow$  importer

**Modified as suggested.**

P3, L78-81: Zhang et al. [2008] and Verstraeten et al. [2015] already showed that through optimizing  $NO_x$  emissions in China, the simulated O3 over the Pacific and over the western US indeed improved.

**We changed this sentence to:**

"Optimization of  $NO_x$  emissions in the upwind regions can improve remote ozone simulations in downwind regions after transport of intercontinental pollution plumes from the free troposphere to the surface [Zhang et al., 2008; Verstraeten et al., 2015]."

Section 2.1 It is unclear in this manuscript how the adjoint accounts for (a) vertical sensitivity of the satellite retrievals, and (b) the diurnal cycle of  $NO_x$  emissions. These aspects are important enough to describe in the manuscript, for (a) useful information is provided in the supplement but it is not clear whether the replacing of the a priori profiles by GEOS-Chem prior profiles was also applied in the research to estimate the emissions. The authors should clarify this in section 2, and also briefly quantify to what extent the differences in prior simulations have been minimized by this approach.

Many of these aspects have been described in details in a previous publication cited in Section 2.2 (Qu et al. 2017), so we do not repeat the same information in this manuscript. To clarify, we added a brief summary of our inversion in Section 2.2, see our response above.

For the reviewer's information, The comparison of SCDs  $(VCD_{GC}AMF_{GC} - SCD_{OMI})$  is theoretically equivalent to comparisons of VCDs  $(VCD_{GC} - \frac{SCD_{OMI}}{AMF_{GC}})$ . These have been described in Qu et al. [2017], pasted below:

"In all of our simulations, we calculate the air mass factor (AMF) for GEOS-Chem simulated NO2 columns ( $AMF_{GC}$ ) following Equations 1 to Equation 4 in Bucsela et al. [2013]. Here,  $AMF_{GC}$  is expressed as the ratio of the sum of slant sub-columns in the troposphere (S) to the sum of vertical sub-columns in the troposphere (V):

where

$$S = \sum_{\substack{l \text{ in the troposphere} \\ l \text{ in the troposphere}}} MR(i, j, l)(P(i, j, l) - P(i, j, l+1))SCW_{OMI}(i, j, l)}$$
$$V = \sum_{\substack{l \text{ in the troposphere} \\ l \text{ in the troposphere}}} MR(i, j, l)(P(i, j, l) - P(i, j, l+1))$$

 $AMF_{GC}(i,j) = \frac{S}{V}$

Here, MR is the mixing ratio of NO2, P is the pressure at the center of the GEOS-Chem grid,  $SCW_{OMI}$  is the scattering weight linearly interpolated from the OMI product to GEOS-Chem grid using the scattering weight pressure from the Level 2 product and pressure at the center of each model grid cell, with application of temperature correction following Equation 4 of Buscela et al. [2013].  $AMF_{GC}$  is then used for conversion of GEOS-Chem NO2 vertical column densities to SCDs, which are directly comparable to SCDs retrieved from OMI,

$$SCD_{GC}(i,j) = AMF_{GC}(i,j) \sum_{l \text{ in the troposphere}} (c(i,j,l) \times h(i,j,l))$$

where c is simulated NO2 concentration [molecules cm-3] and h is the height of the box."

We added the following sentence to the first paragraph of Section 3:

"The cost function has reduced by 6% - 29% in the monthly inversion."

For (b), some info is given but only late in the game (P7: The diurnal variations of  $NO_x$  emission are constrained to be those of the prior emissions), and we do not learn what the diurnal cycle is in the first place. Please revise section 2 thoroughly with this in mind.

We added the following sentence to Section 2.1:

"The diurnal variation of NOx emissions is derived from EDGAR hourly variations (http://wiki.seas.harvard.edu/geoschem/index.php/Scale\_factors\_for\_anthropogenic\_emissions#Diurnal\_Variation) and is not optimized in the inversion."

Then I have other questions:

- how does the adjoint approach account for other relevant aspects of data assimilation?

Details of our 4D-Var inversion are in Qu et al. [2017]. In brief, we define a cost function as described in Section 3 of Qu et al. [2017]. Then, "We minimize the cost function using the quasi-Newton L-BFGS-B gradient-based optimization technique [Byrd et al., 1995; Zhuetal., 1994], in which the gradient of the cost function  $J(\sigma)$  with respect to the control parameter  $\sigma$  is calculated using the adjoint method. The adjoint model is driven by a forcing term, which is the error weighted difference between predicted and simulated NO2 slant columns. Inversions are considered to have converged when the cost function decreases by less than 1% in three consecutive iterations."

- how is the OMI data averaged spatially to the grid of GEOS-Chem, and how are superobservation errors incorporated?

We did not average OMI data or use super-observations. Instead, we assimilate each OMI retrieval separately and compare it with GEOS-Chem simulations at the corresponding hour, with corresponding averaging kernel applied. Please see Section 3 in Qu et al. [2017] for more details, which state:

"Slant column densities from OMI at each observation time and site are used to constrain monthly anthropogenic NOx emissions. The observation error covariance matrix, **S**obs, is assumed to be diagonal. Absolute uncertainties of these diagonal values are read from NASA OMNO2 L2 products for each individual OMI observation. On average, the tropospheric slant column uncertainty of OMI is estimated to be ~ $0.7 \times 10^{15}$  molecules cm-2 [Boersma et al., 2008; Castellanos and Boersma, 2012]. To reduce the influence of observations below the OMI detection limit, which mainly occur in remote locations, we conservatively assume an absolute uncertainty of  $1.0 \times 10^{15}$ molecules cm-2, and we add this value to Sobs."

- did the authors only use the mostly cloud-free OMI retrievals?

Yes, only retrievals with cloud fraction less than 0.2 are used. This has been stated in section 2.2 of this manuscript:

"We screen all OMI NO2 retrievals using data quality flags and by the criteria of positive tropospheric column, cloud fraction < 0.2, solar zenith angle  $< 75^{\circ}$ , and viewing zenith angle  $< 65^{\circ}$ ."

Section 2.2: OMI is suffering from the so-called row anomaly, which was absent until mid-2007, and then became gradually more important. How did the authors ensure that the growing impact of the row anomaly did not unduly affect their trends in NOx emissions?

The OMI data affected by row anomaly are filtered out using the quality flag. We added the following sentences to section 2.2:

**"We excluded all retrievals that are affected by row anomaly."**

We have tested the differences between annual mean OMI NO2 column densities without data filling after excluding pixels affected by row anomaly and when filling missing data by linearly interpolating column densities from adjacent years in Qu et al. [2017]; we found the filling to impact annual mean SCDs by less than 10% for all regions shown in Figure 8 of Qu et al. [2017]. Differences in these two SCDs for all studied years are less than 1% in mainland China.

Another approach to mitigate inconsistent sampling of the data is to follow Duncan et al. [2013] and consider the trend in NO2 columns from only rows 10 to 23 of the NASA standard product, which are unaffected by the row anomaly throughout the period. These are shown in the grey lines in Figure 8 of Qu et al. [2017]. Please also note that even though we are using the same rows each year, this doesn't necessarily mean that the number of observations is the same after screening according to our other filtering criteria, nor does it mean the same geographical locations are observed throughout the period. The correlation of this dataset with OMI data from the standard NASA product in all rows is >0.75 in most regions.

Though we recognize the benefits of using a consistent number of observations to analyze the trend of NO2 columns alone, this is not necessarily the case for a Bayesian inversion of NOx emissions. The inversion is forced by the residual model error summed over all available observations; fewer observations in some years or locations will thus naturally result in greater dependence on the prior emissions. If we exclude observations to maintain consistency in the rows used, emissions in many grid cells do not get updated due to lack of observations (see Fig. R1). This would lead to spatial trends in posterior emissions that could have been avoided if using all available observations (after data screening).

We think the two approaches to invert  $NO_x$  emissions, maintaining consistency in rows used or not, both have their pros and cons. Since the goal of this work is to derive top-down emissions, which would benefit from broader observation coverage (in the example of January 2006 below, we would not be able to get posterior emissions for regions covered in white if eliminating those rows affected by row anomaly throughout) and the trend of  $NO_2$  columns between these two does not differ much, we chose to use all observations available after data selection.

Figure R1. Data coverage in January, 2006, using only rows 10 to 23 (left) and all rows (right), where, red color stands for grid cells that have at least one observation during the month.

Section 2.3: it remains unclear what type of surface station was used for the GEOSChem surface evaluation. Using urban background and regional stations seems appropriate to evaluate the large GC grid cells, but urban street stations should be excluded.

We checked the monitoring site lists and a document defining the site category (http://www.bjmemc.com.cn/xgzs\_getOneInfo.action?infoID=1661). None of the sites included in this study was listed as roadway sites. We added the following sentence to Section 2.3:

"The city monitoring sites included in the analysis represent either urban background or the averaged pollutant concentrations over the city."

P5, L152-154: what explains the OMI-driven differences between the posterior  $NO_x$  emissions, differences in tropospheric slant columns or in the AMFs? Presumably the latter, but since the a priori profile differences have been "minimized", the differences must be in the assumptions on surface albedo and clouds. It would be best if the authors could shed more light on how the scattering weights or averaging kernels are different between the OMI  $NO_2$  retrievals. Please clarify.

We added a new Figure S2 to the supporting information:

---

## Author Comment (AC2) · 8 Aug 2020

Response to review 2

We have responded to each comment below. Our replies are in blue, and the revised manuscript text is written in bold.

This manuscript has presented top-down estimates of global $NO_x$ emissions using two OMI satellite $NO_2$ products over 2005-2016 and using the GEOS-Chem adjoint inversion method. Considerable differences are found between the two top-down emission estimates. Implementing the top-down $NO_x$ emissions to the GEOS-Chem atmospheric chemistry model shows some improvements on the model simulation of tropospheric ozone. The study also points out that model improvements largely depend on the top-down emissions, the ozone metrics used, and model versions. The manuscript is in general well organized and meets the scope of ACP. One main concern is that the manuscript has been presented as a model evaluation paper that comparing several model simulations with different $NO_x$ emissions with surface and sonde ozone measurements. It lacks some analyses in depth to understand the driving factors of the differences. The key findings of this study are also not clear. Do we have a better understanding of the $NO_x$ emission trends as constrained by the satellite measurements, or how $NO_x$ emission changes affect tropospheric ozone? I think the concern and the following specific comments should be addressed before considering publish.

We appreciate the comments from the reviewer. We added the following sentences in the abstract to address the concern on the $NO_x$ emission trends:

**"Posterior $NO_x$ emissions show consistent trend over China, US, India, and Mexico constrained by the two retrievals. Emission trends are less robust over South America, Australia, Western Europe and Africa, where the two retrievals show less consistency."**

Limited by the availability of surface measurements, we cannot claim that $NO_x$ emission trends are improved everywhere. However, we demonstrate in this study that there are several regions where top-down $NO_x$ emission trends are consistent across different retrievals and we are more confident about these.

The impact of $NO_x$ emission on ozone simulations have spatial heterogeneity due to the nonlinear response of ozone to $NO_x$ and our different understanding of local sources, physics, and chemistry. So, there is no generalized conclusion at global scale. We added the following sentences to the abstract to summarize our findings from this work:

**"The performance of posterior ozone simulations is spatially heterogeneous from region to region. On a global scale, ozone simulations using NASA-based emissions remove the double peak in the prior simulation of global ozone. The higher abundances of $NO_2$ from the DOMINO posterior increase the global background ozone concentrations and therefore reduce the negative biases more than the NASA posterior in the GEOS-Chem v12 simulations at remote sites. Compared to surface ozone measurements, posterior simulations have more consistent magnitude and interannual variations than the prior, but the performance from the NASA-based and DOMINO-based emissions varies across ozone metrics. The current hard-constraints on $NO_x$ diurnal variations and limited availability of**

**remote sensing data hinder improvement of ozone diurnal variations from the assimilation, and therefore have mixed performance on improving different ozone metrics. Additional improvements in posterior $NO_2$ and ozone simulations require more precise and consistent $NO_2$ retrieval products, more accurate diurnal variations of $NO_x$ and VOC emissions, and improved simulations of ozone chemistry and depositions."**

Specific comments:

1) Page 1, Line 24-25 in the Abstract: The statement "using $NO_x$ emission datasets that have the best performance . . ." is not clear. As ozone simulation is affected by many other factors, the $NO_x$ emissions that have the best performance on ozone simulation may not be the correct one. Some results in this study also showed that satellite constrained $NO_x$ emissions did not necessarily improve ozone simulation (e.g., China daytime surface ozone in Figure 5)

We have revised the abstract, see response above.

2) Page 3, Section 2.1: What was the spin-up time for the model simulations? Were you using the same initial conditions? Please clarify.

The initial conditions are different for each $NO_x$ emission datasets. We added the following sentence to the last paragraph of Section 2.1:

**"For each $NO_x$ emission dataset, the model is spun-up for 6 months, starting from July 2005. Therefore, we derive $NO_x$ emissions from 2005, but only evaluate simulations with measurements from 2006."**

3) Page 6, Line 179: Should here "the average of GEOS-Chem simulated $NO_2$ column density" be OMI observed $NO_2$ column density over 2x2.5 grid cell? Here you are generating pseudo measurements in the statement. The ratio should be calculated by OMI observations to avoid the OMI vs. model biases.

Thanks for pointing this out. We are calculating in the way the reviewer suggested, but did not describe it correctly. We changed the sentence to:

"…by the ratio of OMI $NO_2$ column density gridded at $0.1° × 0.1°$ to the **OMI $NO_2$ column density gridded at** $2° × 2.5°$ grid cell"

4) Page 8, Line 240-245: The large differences in seasonal variations of DOMINO and NASA posterior $NO_x$ emissions seem interesting. Here you explained that the DOMINO posterior may better constrain soil emissions. Do you have any evidence or support for that?

We changed the cited sentence to:

"The peak of the DOMINO posterior $NO_x$ emissions in the United States and Mexico shifted earlier in the year to June and July compared to the prior and NASA posterior emissions, similar to the results from Miyazaki et al. [2017]. **The peak in DOMINO posterior emissions corresponds to the time of high soil $NO_x$ emissions, which are reported to be underestimated in high-temperature agricultural systems in the bottom-up inventory [Oikawa et al., 2015; Miyazaki et al., 2017].**

5) Page 8, Line 250-256: Here you showed that prior simulated surface ozone concentrations had double maxima in April and August, and the posterior results partly corrected the biases. What cause the double maxima in the prior simulation? And how $NO_x$ emission changes correct the August maximum? Please clarify.

We added the following sentences to the cited paragraph:

**"The August ozone peak in the prior simulation is mainly due to the high ozone concentrations in North China, Southwest China, and North India. The NASA and DOMINO posterior simulations have both reduced surface ozone concentrations in North China Plain and Northeast China in August due to the larger posterior $NO_x$ emissions than the prior in these high-$NO_x$ regions. Both posterior ozone simulations are also smaller than the prior in Tibet and North India due to the reductions of posterior $NO_x$ emissions in low-$NO_x$ region. The August ozone peak in the DOMINO posterior comes from the higher ozone concentrations in Angola and Democratic Republic of the Congo compared to the NASA posterior and prior simulations in the same month and DOMINO posterior simulations in the previous months. This can be explained by the larger upward adjustment of DOMINO posterior $NO_x$ emissions in South Africa in August. These results show the large spatial heterogeneities on the responses of ozone seasonality to the changes in $NO_2$ abundances on a global scale."**

6) Page 9, Line 269-271: As indicated in Figure 6, interannual changes in the two posterior $NO_x$ emissions in Australia over 2005-2016 are not that consistent. The DOMINO results show large reduction over 2006-2010 and then increase afterwards. Do you have any explanation why the two satellite products show different interannual variation and trends over some regions?

The different trends in posterior $NO_x$ emissions are propagated from the different trend in $NO_2$ column densities retrieved from these two products, as shown in Figure R4. This could possibly be caused by the differences in scattering weight / averaging kernel, but it is hard for us to pinpoint what is the exact cause. We made the following modification to the cited sentence:

"In **Mexico**, the two posterior $NO_x$ emissions consistently increased by 6% (NASA) and 13% (DOMINO) from 2005 to 2016. **The DOMINO posterior shows more obvious increase** in Mexico from 2010 to 2016. ... **In Australia, the NASA posterior increases by 10% from 2005 to 2016. In comparison, the DOMINO posterior decreases from 2005 to 2010 and increases afterwards, consistent with the posterior trend from Miyazaki et al. [2017]. The different trends in posterior $NO_x$ emissions are propagated from the trends in the two OMI $NO_2$**

**retrieval products. The discrepancies are likely due to the different surface albedo and cloud products used in the two retrievals, which affect averaging kernel sensitivities.**"

[Figure]

Figure R4. NO₂ column densities in Australia from OMI.

7) Page 10, Line 319: "Ozone measurements in 2014 decreased compared to the 2006 level in China, the US, South America and Mexico". I do not see from Figure 9 that in China ozone concentration in 2014 was lower than 2006.

Thanks for pointing this out. That statement comes from an earlier analysis that used all available TOAR sites at each year, not just sites that have continuous measurements throughout the years. We removed China and South America from that sentence.

8) Page 10, Line 314-316: How did you separate the ozone trends caused by $NO_x$ emissions vs. meteorology? A description in the main text is needed. Also, you may calculate the meteorology (non-$NO_x$) effects using either GC-adj or GCv12 results? Which one did you use in Figure 9, and how they differed?

We added the following sentences to this paragraph:

"**The second trend is calculated through simulations that use constant NO$_x$ emissions throughout the studied years. It has a similar trend from GCv12 and GC-adj as shown in the green lines in Fig. 9. The trend caused by NO$_x$ emissions is obtained by subtracting the second trend from the ozone trend simulated using NO$_x$ emissions at each corresponding year.**"

We also added dotted green lines in Fig. 9 to separately show simulated trend from non-NO$_x$ sources from GCv12 and GCadj.

[Figure]

**Figure 9. Changes of regional mean annual MDA8 ozone concentrations compared to 2006 from TOAR measurements (magenta line), due to changes in bottom-up NO$_x$ emissions (black), due to changes in top-down NO$_x$ emissions (blue lines for simulations from GC-adj and red lines for simulations from GCv12), and due to changes in meteorology and non-**

NO$_x$ emissions (green lines). Only sites that have continuous measurements throughout the 9 years are included. The vertical bars represent the spread of changes from simulations using the NASA and the DOMINO posterior NO$_x$ emissions. The impact of meteorology and natural sources are removed from black, blue and red lines by subtracting simulations using 2010 bottom-up anthropogenic emissions for all years from simulations that use bottom-up NO$_x$ emissions corresponding to each year.

9) Page 11, Line 338: It is surprising that the model versions (GCadj and GCv12) simulate very different ozone vertical profiles. GCv12, which is a more updated version, has much large biases in the upper troposphere, in particular with the updated $NO_x$ emissions. Can you explain why in GCv12 changes in surface $NO_x$ emissions would lead to large ozone changes in the upper troposphere?

GCv12 includes halogen chemistry, which is not included in GCadj. This chemistry depletes ozone. Its impact is especially larger at locations away from $NO_x$ sources, e.g., upper troposphere, leading to much lower ozone concentrations in the GCv12 simulations.

We modified the following sentences in the cited paragraph:

"The different biases in ozone simulations **close to surface** can be explained by the usage of different emission inventories (e.g., different biogenic emissions) **and** different boundary layer mixing scheme (non-local mixing [Lin and McElroy, 2010] in GCv12 and full mixing in GCadj). **The different** chemical **mechanisms in the two model versions affect the different model biases especially in the upper troposphere**. For instance, …"

We also added the following sentences to the cited paragraph:

"**The prior simulations in GCv12 applies $NO_x$ emissions at different altitude, whereas the posterior GCv12 and all GC-adj simulations apply all $NO_x$ emissions to the surface. This leads to different transport and formation of ozone at different model layers and therefore causes larger differences in ozone simulations in the upper troposphere.**"

---

## Author Response (AR2)

Dear editor,

Thanks for the feedback on this work. We have responded to each comment below. Our replies are in blue, and the revised manuscript text is written in bold.

Comments to the Author:
The revised manuscript has improved. The authors provided more insight into the causes for the differences between the NOx emission inversions, and how these differences lead to differences in simulations of tropospheric ozone.

The technical clarifications are much appreciated: in particular the case for using all available OMI measurements in a data assimilation scheme, rather than sampling rows that were outside the row anomaly throughout the entire OMI mission, has been well-argued (Figure R1).

The discussion of GEOS-Chem SCDs is also useful. The finding that GEOS-Chem NO2 SCD generated with the NASA scattering weights exceed the GEOS-Chem SCDs generated with their DOMINO counterpart, is important. It indicates structural differences in the presumptions about the satellite sensitivity to NO2 in the lower troposphere: this is presumed to be stronger in the NASA retrieval than in the DOMINO retrieval. This finding could and should be highlighted more in the final paper.

Thanks for the comments. We added the following sentence to the abstract:

**"The different vertical sensitivities in the two NO2 retrievals affect both magnitude and seasonal variations of top-down NOx emissions."**

We also added the following sentence to the discussion and conclusion:

**"Different vertical sensitivities from the two retrievals are a major cause of the discrepancies in the posterior emissions."**

Remaining issues
1. The concepts of scattering weights, averaging kernels, and vertical sensitivity are used too loosely in the manuscript. In the DOAS-formalisms discussed in Palmer et al. [2001] and Eskes and Boersma [2003], and recently summarized in Chance and Martin [2017], a clear distinction is made between scattering weights and averaging kernels.

Scattering weights (w) are related to SCDs (SCD = ∫ w(z) n(z) dz), with ∫ the integral sign, n(z) the a priori NO2 profile. But averaging kernels (a) are related to VCDs (VCD = ∫ a(z) n(z) dz). Their relationship is via the AMF: a(z) = w(z)/M, with M the AMF. I recommend to first clearly define, and then carefully check every use of the term 'scattering weights', 'averaging kernels', and 'vertical sensitivity', in the manuscript. This is important in order to prevent the wrong use of these concepts.

We changed the first sentence in the second paragraph of Section 2.2 to:

"We converted GEOS-Chem $NO_2$ VCD to SCD using scattering weight from the OMI retrievals and then compare GEOS-Chem SCD with SCD retrieved from OMI. **The scattering weights are the product of the averaging kernels and the air mass factor (AMF) [Palmer et al., 2001; Chance and Martin 2017].**"

We also changed "averaging kernel" to "**scattering weight**" on line 281.

2. It remains unclear how the uncertainty in the AMF (the observation operator in generating the GEOS-Chem SCDs) is accounted for in the assimilations. This is important because, together with the estimated uncertainty on the model state, it determines how strongly OMI is driving the data assimilation. In other words, I suggest the authors provide the relative weight of the OMI SCD vs. the GEOS-Chem SCD in the assimilation scheme.

We did not explicitly include uncertainties in the AMF. While these are not provided in all of the retrievals we used, and it is beyond the scope of this paper to calculate them, based on those that were provided for the NASA standard product in January 2010 the relative contribution of AMF uncertainty to the uncertainty of the tropospheric NO2 SCD is ~2%.

3. I strongly disagree with the phrase that "daily NO2 column densities from OMI are underestimated to the diurnally varying ground-based retrievals [Herman et al., 2019]." As stated in my previous review, OMI is simply measuring at 13:30 hrs, close to the diurnal minimum in NO2 columns. To then call this an "underestimate" is misleading.

We changed the cited sentence to "The daily NO2 column densities from OMI are **smaller** compared to the diurnally varying ground-based retrievals"

4. L26: please clarify what is meant with "current hard-constraints on NOx diurnal variation". There is no constraint from the satellite data, so in essence the prior diurnal variation is used. Please rephrase to make this clear.

"Hard-constraint" or "strong-constraint" as opposed to "weak-constraint" within the 4D-Var framework, i.e., an aspect of the model that is not adjusted. But this is jargon from the assimilation community, so to clarify we changed the cited sentence to:

[revised manuscript text omitted]